# ADAMTS4-specific MR probe to assess aortic aneurysms in vivo using synthetic peptide libraries

Jan O. Kaufmann[1,2,3,19], Julia Brangsch[1,4,19], Avan Kader [1,5,6], Jessica Saatz[7], Dilyana B. Mangarova[1,8], Martin Zacharias [9], Wolfgang E. Kempf[10], Timm Schwaar[11], Marco Ponader[3], Lisa C. Adams [1], Jana Möckel[1], Rene M. Botnar [12,13,14,15,16,17,18], Matthias Taupitz[1], Lars Mägdefessel[10], Heike Traub [7], Bernd Hamm[1], Michael G. Weller [3] & Marcus R. Makowski [1,6,12,17✉]

The incidence of abdominal aortic aneurysms (AAAs) has substantially increased during the last 20 years and their rupture remains the third most common cause of sudden death in the cardiovascular field after myocardial infarction and stroke. The only established clinical parameter to assess AAAs is based on the aneurysm size. Novel biomarkers are needed to improve the assessment of the risk of rupture. ADAMTS4 (A Disintegrin And Metalloproteinase with ThromboSpondin motifs 4) is a strongly upregulated proteoglycan cleaving enzyme in the unstable course of AAAs. In the screening of a one-bead-one-compound library against ADAMTS4, a low-molecular-weight cyclic peptide is discovered with favorable properties for in vivo molecular magnetic resonance imaging applications. After identification and characterization, it's potential is evaluated in an AAA mouse model. The ADAMTS4-specific probe enables the in vivo imaging-based prediction of aneurysm expansion and rupture.

[1] Charité – Universitätsmedizin Berlin, corporate member of Freie Universität Berlin, Humboldt-Universität zu Berlin, and Berlin Institute of Health, Charitéplatz 1, 10117 Berlin, Germany. [2] Humboldt-Universität zu Berlin, Department of Chemistry, Brook-Taylor-Str. 2, 12489 Berlin, Germany. [3] Federal Institute for Materials Research and Testing (BAM), Division 1.5 Protein Analysis, Richard-Willstätter-Str. 11, 12489 Berlin, Germany. [4] Institute of Animal Welfare, Animal Behavior and Laboratory Animal Science, Freie Universität Berlin, Königsweg 67, Building 21, 14163 Berlin, Germany. [5] Institute of Biology, Freie Universität Berlin, Königin-Luise-Str. 1-3, 14195 Berlin, Germany. [6] Department of Radiology, Klinikum rechts der Isar, Technische Universität München (TUM), Ismaninger Straße 22, 81675 Munich, Germany. [7] Federal Institute for Materials Research and Testing (BAM), Division 1.1 Inorganic Trace Analysis, Richard-Willstätter-Str. 11, 12489 Berlin, Germany. [8] Institute of Veterinary Pathology, Freie Universität Berlin, Robert-von-Ostertag-Str. 15, Building 12, 14163 Berlin, Germany. [9] Center of Functional Protein Assemblies, Technische Universität München (TUM), Ernst-Otto-Fischer-Str. 9, 85748 Garching, Germany. [10] Department for Vascular and Endovascular Surgery, Klinikum rechts der Isar, Technische Universität München (TUM), 81675 Munich, Germany. [11] Federal Institute for Materials Research and Testing (BAM), Division 1.0 SAFIA Technologies, Richard-Willstätter-Str. 11, 12489 Berlin, Germany. [12] King's College London, School of Biomedical Engineering and Imaging Sciences, London, UK. [13] Wellcome Trust / EPSRC Centre for Medical Engineering, King's College London, London, UK. [14] BHF Centre of Excellence, King's College London, London, UK. [15] Escuela de Ingeniería, Pontificia Universidad Católica de Chile, Santiago, Chile. [16] Millennium Institute in Intelligent Healthcare Engineering, Santiago de Chile, Campus San Joaquín - Avda.Vicuña Mackenna, 4860 Macul Santiago, Chile. [17] St Thomas' Hospital Westminster Bridge Road, London SE1 7EH, UK. [18] Denmark Hill Campus, 125 Coldharbour Lane, London SE5 9NU, UK. [19]These authors contributed equally: Jan O. Kaufmann, Julia Brangsch. ✉email: marcus.makowski@tum.de

Cardiovascular disease remains the leading cause of death, and aortic disease accounts for an increasing part of the overall disease burden[1]. The incidence of abdominal aortic aneurysms (AAAs) has increased substantially in the last 20 years[2,3]. Fatal rupture of AAAs is the third most common cause of sudden death after myocardial infarction and stroke in the field of cardiovascular diseases. There is an estimated incidence of AAAs of ~5% in populations over 50 years of age[4]. Most AAAs are still classified as non-specific, as in most cases, no defining event for the formation of AAAs can be identified[5]. Independent of the cause, rupture of the aortic wall usually leads to a progressive dilatation and, if unrecognized or untreated, to potential fatal aortic rupture[6].

In the clinical setting, AAAs are diagnosed by the screening of the abdominal aorta using ultrasound, computed tomography angiography (CTA), or magnetic resonance angiography (MRA). The majority of the identified AAAs are small in size and do not need immediate surgical intervention[7,8]. However, AAAs can enlarge over time, and their risk of rupture increases with the expansion of their diameter[9–11]. There is common consent that patients with AAAs larger than 55 mm should undergo surgery[6]. There is however still controversy regarding the management of asymptomatic medium-sized AAAs (40–55 mm)[6,8]. Currently, there is no established biomarker available, which could improve the characterization of AAAs and predict their risk of rupture[6,8].

Small peptides are promising candidates for the development of molecular magnetic resonance imaging (MRI) probes due to their favorable distribution in the blood system, their rapid blood clearance through the renal system, and their specific binding properties[12–15]. Among several different combinatorial screening methods, one-bead-one-compound libraries (OBOC) are particularly well suited for the screening of small cyclic peptide binders[15–17]. By combining the screening and sequencing part on the same chip, potential drawbacks of this method, including manual separation of positive hits, can be avoided[18].

The extracellular matrix enzyme ADAMTS4 (A Disintegrin And Metalloproteinase with ThromboSpondin motifs 4) is strongly upregulated in cardiovascular diseases, including aneurysm and atherosclerosis, and therefore represents a promising in vivo target[19–21]. ADAMTS4's expression is specifically linked to fatal aortic rupture[22].

Here we show, that by screening a fully synthetic OBOC library we were able to identify a selective binder against ADAMTS4. Subsequently, we transformed the peptide into a molecular MR probe for the characterization of AAAs in vivo.

## Results

**Library synthesis and screening.** In the first step, we set up a system for the identification of specific small MR peptides which can be used for imaging. We adapted the applied on-chip screening technique to the requirements of in vivo MR imaging with an emphasis on cyclic peptides. Due to the more rigid structure, cyclic peptides have large surface areas and decreased entropy terms of the Gibbs free energy, which favored high and selective binding. An additional favorable characteristic of the cyclic structure is the higher hydrolysis stability against protease activities compared to linear peptides[23,24]. We focused on a cycle containing four fully randomized amino acids out of the eleven canonical amino acids, framed by two cysteines to form a cycle by a disulfide bridge (Fig. 1a). For the eleven canonical amino acids, we used each kind of amino acid at least once. This included a hydrophobic, hydrophilic, basic, acetic, and aromatic amino acid. Cysteine and methionine were excluded to avoid unintended disulfide formation by additional cysteine and unintended oxidation of methionine[25]. The cyclic peptide was linked by a small

spacer sequence (Gly-Arg-Ser-TTDS-TTDS-Ser) to a base-labile HMBA (4-hydroxymethylbenzoic acid) linker. By this construct, the peptide can be cleaved from the resin by ammonia vapor without solubilization in a solvent[26].

We used the truncation technique of St. Hilaire et al.[27]. with 7% of Boc-protected amino acids to the 93% of standard Fmoc-protected amino acids. Therefore, we created a ladder sequence, since every integration of a Boc amino acid stops any further coupling. After the four coupling steps, estimated 75% of the full-length peptide could be presumed. For quality control of the library, we used a peptide with defined sequence, which we synthesized simultaneously to the library as control. With this peptide, we could examine the duration of the cyclisation and the truncation sequence in the matrix-assisted laser desorption/ionization-time of flight-mass spectrometry (MALDI-TOF-MS). Additionally, we developed test-chips with the library and examined them with MALDI-TOF-MS, to test, if we can measure the truncation also under screening conditions. In contrast to the test-peptide, we could not detect the truncation sequences on the chip. Therefore, we tested the alkylation of the free cysteine in the truncation sequence. We used N-ethylmaleimide (NEM) and iodoacetamide (IA) for alkylation. We observed highly double alkylated peptides with NEM and focused more on IA in the MALDI-TOF-MS. By checking four different time points (15 min, 30 min, 45 min, and 60 min) we identified the most promising results after 30 min of IA. We then alkylated the free thiol moiety in the truncation peptides with iodoacetamide, which reveals the truncation sequences in MALDI-TOF-MS.

In the second step, we focused on the screening against ADAMTS4 using our library of cyclic peptides. By using this library in the automated high-throughput screening[18] we were able to screen 22.000 beads from a 300.000 beads library against the metalloprotease ADAMTS4. The screening was performed on a single chip, which was repeatedly regenerated for multi-step incubations with different targets, like the similar ADAMTS5, to reduce false-positive hits and increase selectivity (Fig. 1a). After each screening step, we measured the fluorescence, to check for false-positive hits or broken beads. The screening resulted in several positive hits against ADAMTS4/5 (Fig. 1b). The five most promising peptides were resynthesized. Two candidates showed stronger fluorescence for ADAMTS4 than for ADAMTS5, indicating a higher selectivity. The other three showed comparable fluorescence for ADAMTS4/5 (Supplementary Fig. 1a).

Subsequently, we sequenced the peptides by MALDI-TOF MS, resynthesized them, and examined the five peptides by single-cycle surface plasmon resonance (SPR). We identified one candidate as the most promising binding peptide for ADAMTS4 with the sequence Cys*-Ser-Arg-Arg-Gly-Cys*-Ser (Supplementary Fig. 1b, c and Supplementary Fig. 2). To our knowledge, this sequence has not been reported previously. It shows the potential of the OBOC screening technique for the discovery of cyclic peptide binders, for molecular MR in vivo applications.

**Synthesis, biodistribution, binding, and imaging properties of the ADAMTS4-specific probe.** Subsequently, the peptide was modified for in vivo MR imaging by conjugation of a short peptide linker and a Gd-DOTA complex (Fig. 2a). A polyethylene glycol (PEG) linker used during the screening was replaced with a short peptide linker containing glycine and serine. The flexible and hydrophilic residues of the amino acids reduce interaction with proteins and do not form secondary structures[28–30]. Furthermore, we inserted a Gd-DOTA complex (gadolinium 1,4,7,10-tetraazacyclododecane-1,4,7,10-tetraacetic acid), a Cy5 or a 5-carboxyfluorescein (FAM-5) dye via a lysine side chain. To determine the binding constant of our ADAMTS4-specific probe,

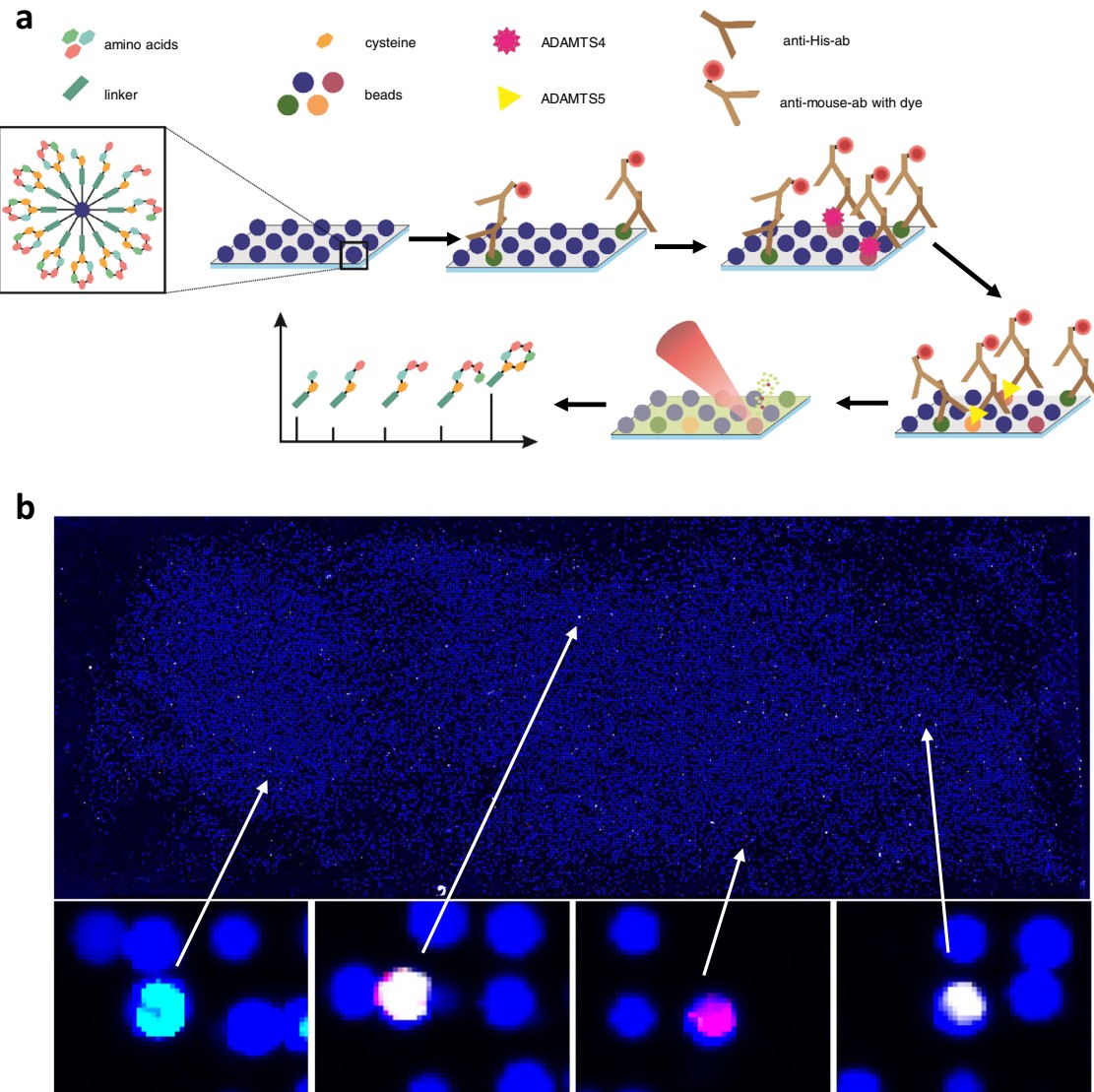

**Fig. 1 Introduction of a one-bead-one-compound-library for screening against ADAMTS4 and evaluation of the binding properties of the discovered low-molecular-weight cyclic highly selective peptide targeted against ADAMTS4. a** A one-bead-one-compound-library was used to screen for binders against ADAMTS4. First, we adapted our on-chip screening technique to the requirements of in vivo MR imaging with an emphasis on cyclic peptides to achieve high peptide stability against proteases and high binding affinities. For the screening, we used our target protein, which was detected by an anti-His-Ab. The anti-His-Ab was afterward stained with an anti-Mouse IgG-Atto 633 antibody. After immobilization of the beads on a chip, firstly, a prescan against both antibodies was performed, to exclude positive hits against the antibodies. This was followed by the main scan against ADAMTS4 and a second scan against ADAMTS5. After each step, the chip was regenerated by guanidine hydrochloride. For MALDI-TOF-MS identification, the peptides were cleaved from the resin with ammonia gas, and a matrix was applied. In the last step, the amino acid sequence can be determined by MALDI-TOF-MS on-chip. **b** During the screening, different types of hits were detected; positive hits against (from left to right) the antibodies (cyan), against ADAMTS4 and ADAMTS5, only ADAMTS4 (magenta), or only ADAMTS5 (white). MALDI-TOF-MS: Matrix-assisted laser desorption/ionization-time of flights-mass spectrometry; MR: magnetic resonance.

we used the Cy5-labeled probe and measured it by microscale thermophoresis (MST) (Fig. 2b). In our case, we detected a ligand-dependent fluorescence decrease (Supplementary Fig. 4a–h), which we confirmed by a specificity test with denaturation with urea. In such cases, the binding affinity must be determined by the fluorescence change, not the MST. We achieved a binding constant of $51 \pm 21$ nM, which is in the ideal range for a molecular MR probe (Fig. 2b and Supplementary Fig. 3)[12,13]. Additionally, we measured the same probe against ADAMTS1 and ADAMTS5 (Supplementary Fig. 3a, b). For both proteins, we can observe a starting binding in a higher concentration range, but we did not reach the plateau. Therefore, a

determination of the binding constant is not possible, but the affinity is at least 20 times weaker as for ADAMTS4. Furthermore, by performing an alanine scan, the binding disappeared by the exchange of the arginine. The exchange of the glycine leads to a decrease in the binding affinity of around ten times to $609 \pm 374$ nM. For the exchange of serine, we could not reach the plateau again, so a determination of the binding constant was not possible (Supplementary Fig. 3a, c). As a negative control, we used one of the peptides from the screening, which showed not only a weak signal against ADAMTS4, but also only a weak interaction in the SPR experiment. The peptide with the sequence Cys*-Phe-His-Pro-Tyr-Cys*-Ser did not show binding in the

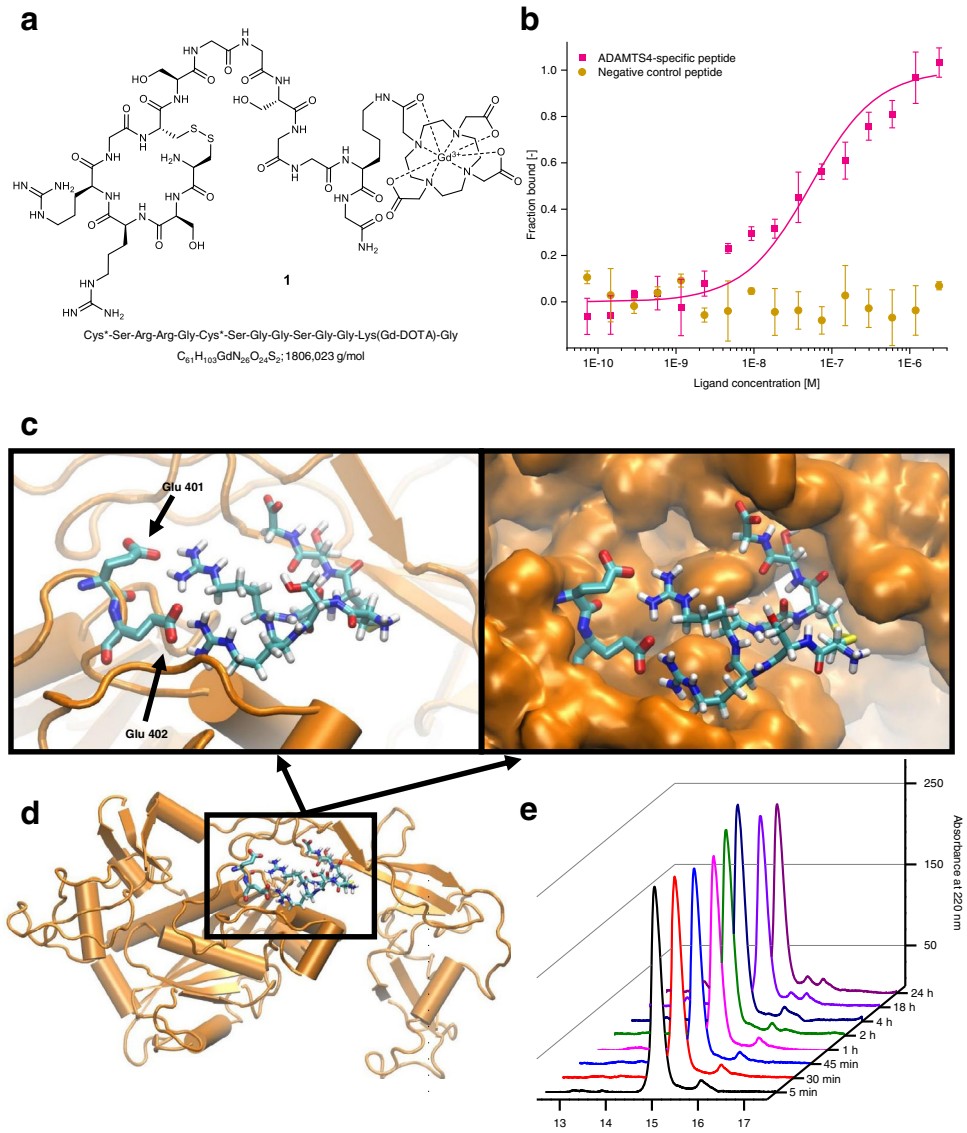

**Fig. 2 Evaluation of the binding affinity (including negative control peptide), binding site, and stability of the ADAMTS4-specific probe. a** The ADAMTS4-specific probe contains 14 amino acids with a small cyclic ring of six amino acids. The MR active DOTA-Gd complex is coupled via a lysine side chain. The binding part and the MR active part are linked by a polar linker of glycine and serine amino acids. **b** By using MST, the binding constant between ADAMTS4 and the Cy5-labeled probe could be determined and showed a medium nanomolar affinity of 51 ± 21 nM, which is in the ideal range for specific MRI probes. However, for the negative control peptide, we could not find binding in this concentration range. As negative control, we used the peptide sequence from the screening, which showed also very low interaction in the SPR experiment with the sequence Cys*-Phe-His-Pro-Tyr-Cys*-Ser (n = 3 biologically independent measurements for each peptide; the data are presented as mean values ± SD). **c** Using molecular docking for predicting putative interaction geometries between a model structure of ADAMTS4 and the ADAMTS4-specific cyclic peptide, a binding cleft in ADAMTS4 was identified, in which the simultaneous contact between two glutamic acid residues from the protein and the two arginine residues of the peptide is indicated. Additionally, the peptide shows high contact to the ADAMTS4 protein (space-filled structure) and no dissociation during an MD simulation was observed. The binding cleft is unique for ADAMTS4 and was not found for ADAMTS1 and ADAMSTS5. **d** The binding cleft is located near to the protein surface and since the shortened linker of the peptide pointed outside, access to water is feasible, which is required for the generation of the T1 signal of the MR probe. **e** No relevant degradation of the ADAMTS4-specific probe within 24 h was measured in the human plasma by HPLC. HPLC High-performance liquid chromatography, MST Microscale thermophoresis, nM nanomolar, KD: binding constant; MRI magnetic resonance imaging. Source data for **b**, **e** are provided as a Source Data file.

ligand-dependent fluorescence change (Fig. 2b). For the three peptides with no binding, we checked also the MST trace, but could not detect a binding either (Supplementary Fig. 4i–k).

We performed a docking experiment with ADAMTS4 from the AlphaFold-Server[31] in the sequence area of Phe213-Cys685. This is consistent with the recombinant proteins, we used in the screening process and the shortened ADAMTS4-binding peptide. In this docking experiment, we identified a potential binding

pocket. The pocket is in a flexible sector and lies above the S1' binding side[32]. The cyclic peptide fits in this pocket with strong interactions with the protein without steric hindrance, and it permits the formation of two salt bridges between the two glutamic acid residues Glu401 and Glu402 from the human ADAMTS4 and the two guanidino groups from the peptide arginine (Fig. 2c). We repeated the docking experiment also with mouse ADAMTS4, in which the two glutamic residues are slightly

shifted to positions 397 and 398. The binding cleft could also be found (Supplementary Fig. 5). By exchange of the glutamic acids by glutamine, the binding was lost (Supplementary Fig. 6a,b). This is consistent with the loss of binding in the alanine scan, in which the arginine of the peptide as ionic binding partners to the glutamic acid residues was exchanged by alanine. By performing a dynamic simulation, the peptide-protein complex was stable during the MD time (please see Supplementary video 1). We repeated the dynamic simulation for three separate simulations of human ADAMTS4 with different initial velocity distributions and one time for the mouse ADAMTS4. All simulations remained stable. For the mutation of the glutamic acids, the complex breaks apart after 8 ns (Supplementary Fig. 6a). Additionally, both glutamic acids in the pocket exist only in ADAMTS4 and not in ADAMTS1 or ADAMTS5. This coincides with the experimental data from the screening and MST measurements, where only weak interaction between the ADAMTS4-specific peptide and ADAMTS4 closely related enzymes ADAMTS1 and ADAMTS5[32] could be observed. Since the linker in the peptide points to the outside of the protein, free accessibility to water and, thus, a high relaxivity is ensured (Fig. 2d). The relaxivity of the bound and unbound probe was evaluated ex vivo at 3 T by applying the probe to ADAMTS4. The probe demonstrated favorable values for MR imaging with a longitudinal relaxivity of the unbound and bound probe of $3.4 \pm 0.19$ mM$^{-1}$ s$^{-1}$ and $6.13 \pm 0.73$ mM$^{-1}$ s$^{-1}$.

The metabolic stability of the ADAMTS4-specific probe in human plasma was tested using high-performance liquid chromatography (HPLC) at different time points ranging from 5 min to 24 h. No relevant biodegradation of the ADAMTS4-specific probe within 24 h was observed (Fig. 2e).

The potential toxicity of the ADAMTS4-specific probe was tested in an MTT assay a CytoTox-One assay and a CellTiter-Glo assay by using mouse and human aortic endothelial cells (Supplementary Fig. 7a, b, c). Thereby, the cells were affected by the probe for 3 h, 24 h, and 48 h. The MTT assay showed no significant difference at all concentrations compared to the negative control ($p > 0.05$; $n = 6$ for each time point, each concentration, each cell line). The analysis with the positive control shows a significant difference ($p < 0.01$). The result could be confirmed with the CellTiter-Glo and CytoTox-One assay. The measurements showed that no significant cell changes are induced by the ADAMTS4-specific probe (Supplementary Fig. 7a-c).

In the next step, we investigated the biodistribution of the probe by quantitative inductively coupled plasma mass spectrometry (ICP-MS) of digested tissue, blood, and urine samples at three different time points in male apolipoprotein-E knockout (ApoE$^{-/-}$) mice ($n = 3$ per time point). Tissue samples of liver, lung, spleen, heart, and muscle showed low gadolinium uptake and rapid Gd clearance after 120 min. Higher levels of Gd were found in the kidneys and urine, indicating the expected renal clearance of the MRI probe (Supplementary Fig. 8a). Blood clearance of the ADAMTS4-specific probe was rapid (Supplementary Fig. 8b).

**In vivo MR imaging using the ADAMTS4-specific probe**. To determine the potential of the ADAMTS4-specific probe in the development and progression of AAAs, MRI was performed in male ApoE$^{-/-}$ mice at 2 and 4 weeks following aneurysm induction (Supplementary Fig. 8a). The infusion with angiotensin II via osmotic minipumps resulted in an increasing aortic dilatation (Fig. 3b, c). Saline-infused mice, serving as the control group, developed no aortic dilatation (Fig. 3a). After 2 weeks of AngII-infusion, a significant increase of 88% ($n = 9$; $P < 0.01$) in MR signal was visible after administration of the ADAMTS4-

specific probe in the area of the abdominal aortic wall (Fig. 3b, d). Moreover, after 4 weeks of AngII-infusion, an increase of 166% in post-contrast MR signal was observed ($n = 10$; $P < 0.01$), associated with a further accumulation of ADAMTS4 within the aneurysmal wall based on ex vivo histology (Fig. 3c, d). In the sham group, a small contrast-to-noise ratio (CNR) increase of 17% due to the circulating contrast agent could be observed ($n = 10$; $P < 0.01$).

A longitudinal sub-study was performed to evaluate the potential of molecular MRI using an ADAMTS4-specific probe as a prognostic tool for AAA progression. 12 ApoE$^{-/-}$ mice ($n = 12$) were imaged with the ADAMTS4-specific probe 1 week after AAA induction and consequently the same group of animals was imaged by native MRI after 4 weeks (Supplementary Fig. 9b). The administration of the ADAMTS4-specific probe in the first week resulted in a significant increase of 88% in MR signal in animals, suffering from an extensive AAA progression as visible in MRI after 4 weeks ($n = 6$; $P < 0.01$). Animals, that did not develop AAAs over the time of 4 weeks ($n = 6$; $P < 0.01$), also showed an increase of CNR of 19%, which is not significant compared to the sham group in the previous in vivo experiment ($P > 0.05$) (Fig. 3e, f).

Additionally, we performed a second longitudinal sub-study, in which we examined the potential of our probe for the early detection of AAAs. Therefore, 20 ApoE$^{-/-}$ mice ($n = 20$) were implanted with angiotensin-loaded minipumps. The mice underwent a native MRI followed by an ADAMTS4-MRI on days 2, 4, and ten after AngII-infusion (Supplementary Fig. 9c). On day 28 post implantation, native imaging was performed to assess whether an AAA developed. Subsequently, the aortic samples were excised. In this longitudinal sub-study setup, we could demonstrate that the ADAMTS4-MRI signal in the aortic wall significantly increases 4 days after the implantation of the AngII-minipumps. Only a minor further increase can be measured after 10 days (Fig. 4a-c).

More importantly, we could demonstrate that animals with the highest increase in ADAMTS4-MRI signal (CNR: $15.2 \pm 0.7$, Fig. 4c) were most likely to suffer a fatal AAA rupture. Animals with a moderate increase in ADAMTS4-MRI signal (CNR: $10.3 \pm 1.1$, Fig. 4b), which developed an AAA survived for 4 weeks. Animals with no significant increase in ADAMTS4-MRI signal did not develop an AAA (CNR: $6.9 \pm 1.2$, Fig. 4a). The 4-day time point, therefore, was the most promising early time point for the prediction of AAA development and rupture.

Based on the longitudinal study setup described above, we could demonstrate that the ADAMTS4-MRI signal is a stronger predictor for the development of a fatal AAA rupture (area under the curve: 0.96), compared to the aortic area (area under the curve: 0.78, Fig. 4e, f).

To test the potential of the probe for in vivo MR imaging, a subsequent ex vivo MRI was performed of three AAA samples. The MR signal enhancement of ex vivo MR imaging of AAA samples was consistent with in vivo MRI and histological analysis (Fig. 5a).

For evaluation of the specific in vivo binding of the ADAMTS4-specific probe, a fluorescein-labeled probe was administered intravenously in one ApoE$^{-/-}$ mouse 4 weeks after AAA induction and one sham-operated mouse. The resulting fluorescence on histological sections was compared with the immunofluorescence by ADAMTS4-antibody staining (Fig. 5b). Both showed fluorescence in similar regions, which supports the specificity of our probe.

To further check for in vivo unspecific binding of the ADAMTS4-specific probe, an in vivo competition experiment was performed in three ApoE$^{-/-}$ mice ($n = 3$) 4 weeks after AAA induction. Injection of a sevenfold higher dose of a non-

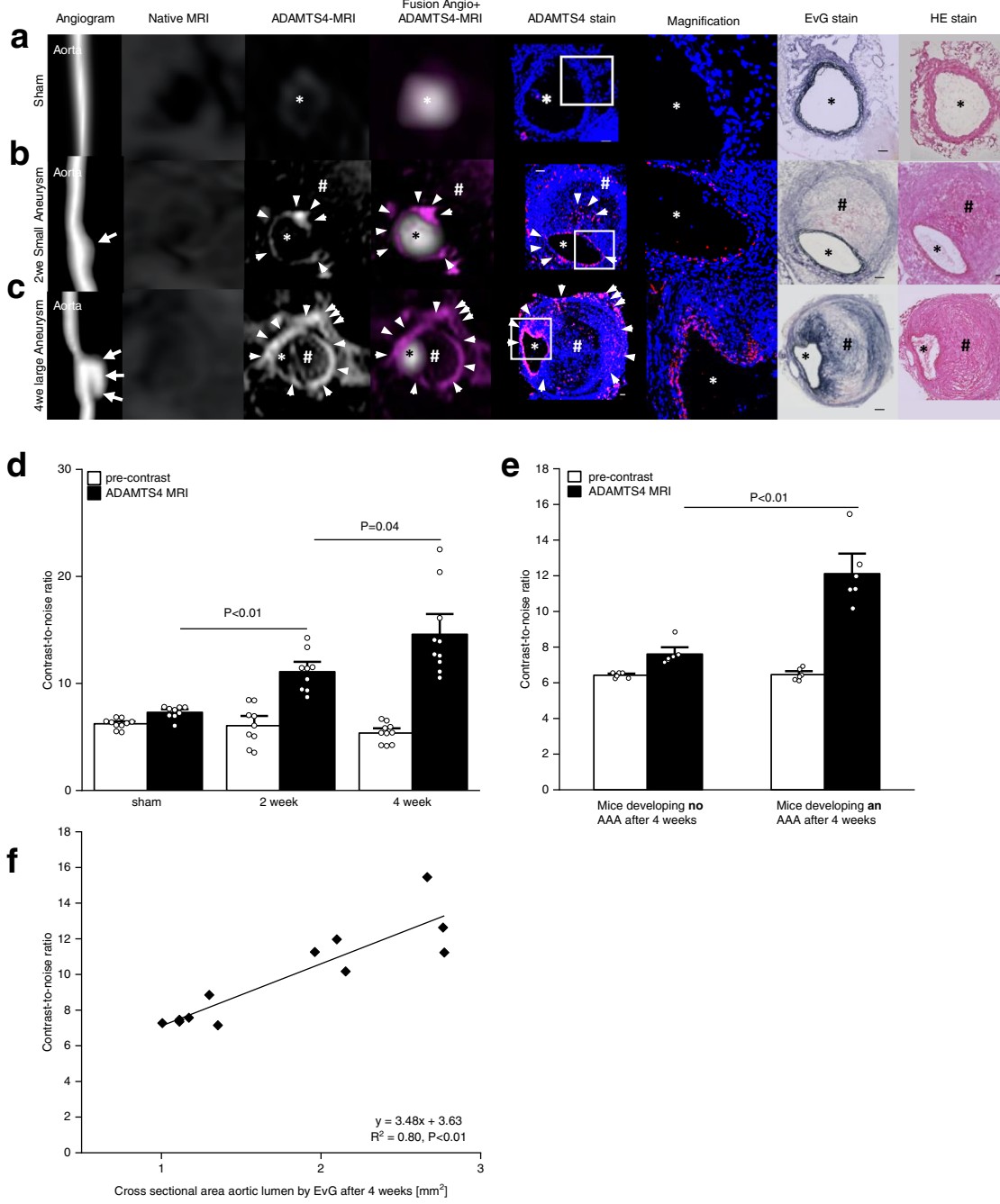

**Fig. 3 In vivo MR imaging using a low-molecular-weight cyclic highly selective peptide targeted against ADAMTS4. a–c** Time-of-flight angiograms show the suprarenal abdominal aorta of male apolipoprotein-E knockout mice (first row). **a** A sham-operated mouse receiving saline showed no pathological changes of the aortic wall in vivo and no ADAMTS4 expression. Two (**b**) and 4 weeks (**c**) following angiotensin-infusion, aortic dilatation (white arrows) and a clear signal enhancement in the aneurysmal wall were observed on T1-weighted sequences after administration of the ADAMTS4-specific probe (arrow-heads). Ex vivo histology confirmed the formation of abdominal aortic aneurysm and a strong expression of ADAMTS4 within the aneurysmal wall (arrow-heads). **d** A significant increase of contrast-to-noise-ratio was observed after administration of the ADAMTS4-specific probe on in vivo MRI scans with a significantly stronger enhancement between sham animals ($n = 10$), the 2 weeks group ($n = 10$) ($P = 0.0001$), and the 4 weeks group ($n = 10$) ($P = 0.04$). Data are presented as mean values ± SEM. **e** Mice of the longitudinal group ($n = 12$) that developed an extensive progression of the abdominal aneurysm after 4 weeks of angiotensin II—infusion showed a significantly higher MR signal enhancement ($P = 0.0008$) after injection of the ADAMTS4-specific probe compared to mice with no further aneurysm progression. Data are presented as mean values ± SEM. **f** In vivo contrast-to-noise ratio measurements after 4 weeks ($n = 12$) correlated well with histological analysis ($y = 3.48x + 3.63$; $R^2 = 0.80$; $P = 1.1E$-09). TOF Time-of-flight, MRI magnetic resonance imaging, EvG Elastica van Gieson staining, HE Hematoxylin-Eosin staining, AAA abdominal aortic aneurysm, ECM extracellular matrix. * indicates the aortic lumen; # indicates the thrombus area; Scale bars represent 100 μm; Source data for **d**, **f** are provided as a Source Data file. For data analysis, a two-sided, unpaired t-test was used.

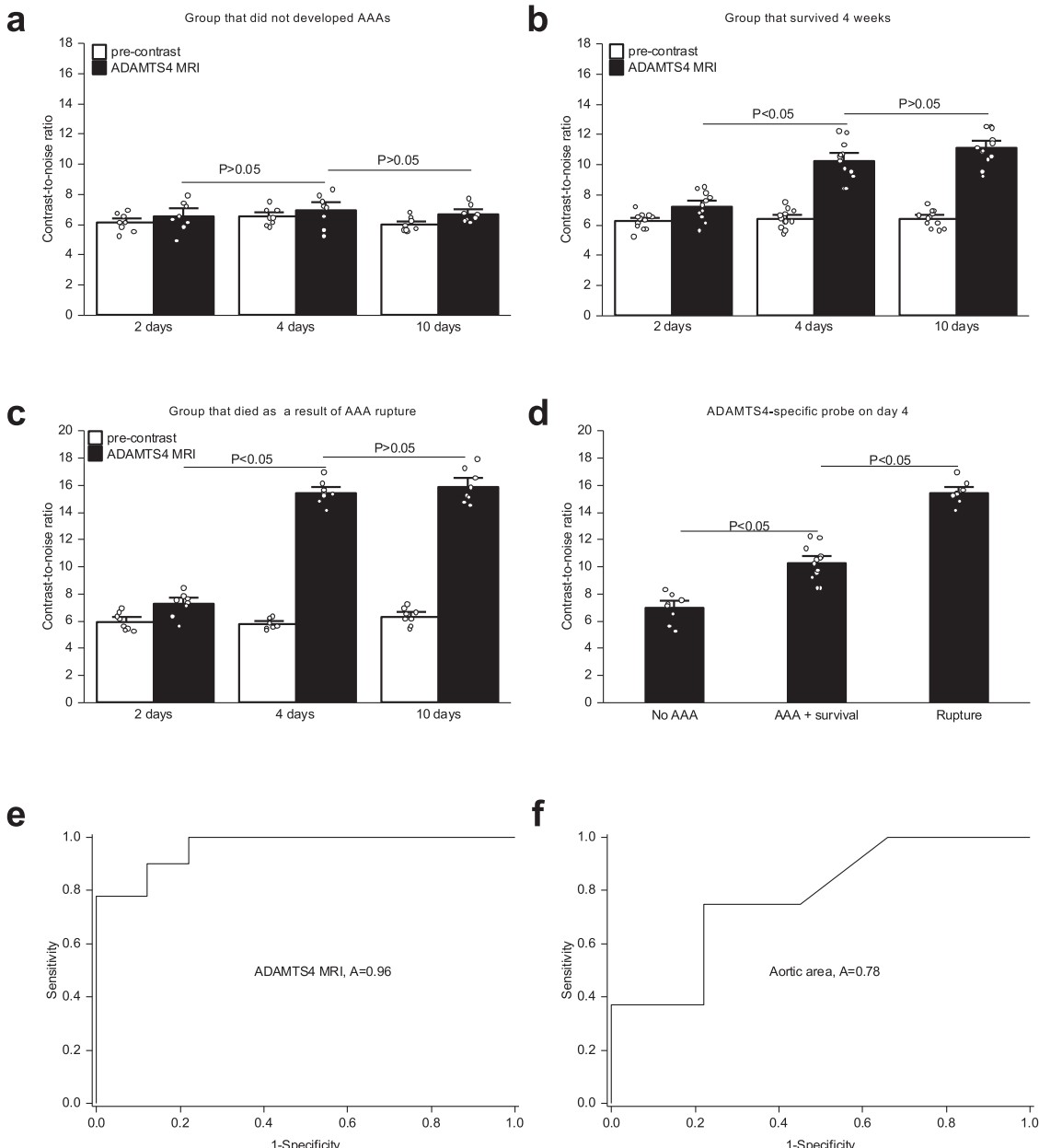

**Fig. 4 Contrast-to-noise ratio for early detection of aneurysm and rupture prediction. a** The group that did not develop an AAA did not show a significant increase in ADAMTS4-MRI levels 2, 4, or 10 days after the initiation of AngII-infusion ($n = 8$, $P = 0.21$; $P = 0.57$). Data are presented as mean values ± SEM. **b** The group ($n = 12$) that developed an AAA however survived for 4 weeks showed a moderate increase of ADAMTS4-MRI levels after 4 days ($P = 9.1E\text{-}06$) with no further significant increase after 10 days ($P = 0.14$). Data are presented as mean values ± SEM. **c** The fatal AAA rupture group ($n = 8$) showed the strongest increase in ADAMTS4-MRI levels of the 4 days ($P = 1.79E\text{-}06$), with no further significant increase after 10 days ($P = 0.19$). Data are presented as mean values ± SEM. **d** ADAMTS4-MRI levels measured after 4 days significantly increased in the group that developed in AAA and survived for 4 weeks, compared to the group that did not develop any AAA ($P = 0.001$). The group that suffered a fatal AAA rupture showed further significant in increase ADAMTS4-MRI levels at this time point ($P = 3.23E\text{-}05$). The 4-day time point therefore seems to be the most promising time point for the prediction of an AAA development and survival. Data are presented as mean values ± SEM. **e** The ROC analysis demonstrates that the ADAMTS4-MRI level measured after 4 days enables of the prediction of a fatal AAA rupture with a sensitivity of 0.88 and a specificity of 0.9 (ROC Curve Area 0.96, Standard Error 0.04, 95% Confidence Interval 0.90 To 1.04). **f** The ROC analysis demonstrates that the aortic area measured after 4 days enables of the prediction of a fatal AAA rupture with a sensitivity of 0.78 and a specificity of 0.76 (ROC Curve Area 0.78, Standard Error 0.11, 95% Confidence Interval 0.55–1.01). AAA abdominal aortic aneurysm, MRI magnetic resonance imaging, AngII angiotensin II, ROC receiver operator characteristic Source data for **a**–**f** are provided as a Source Data file. For data analysis, a two-sided, unpaired *t*-test was used.

paramagnetic europium-labeled ADAMTS4-specific probe resulted in a significant decrease in contrast-to-noise ratio compared to the administration of the gadolinium-labeled ADAMTS4-specific probe alone (Fig. 5c). Since the "cold" probe cannot block any unspecific binding site with only a sevenfold

higher dose, it should be enough to block the ADAMTS4-binding sites. Thus, our probe does not show unspecific binding.

For further evaluation of the specific binding of the ADAMTS4-specific probe, a negative control probe was synthesized and applied for an in vivo experiment in 6 ApoE$^{-/-}$ mice

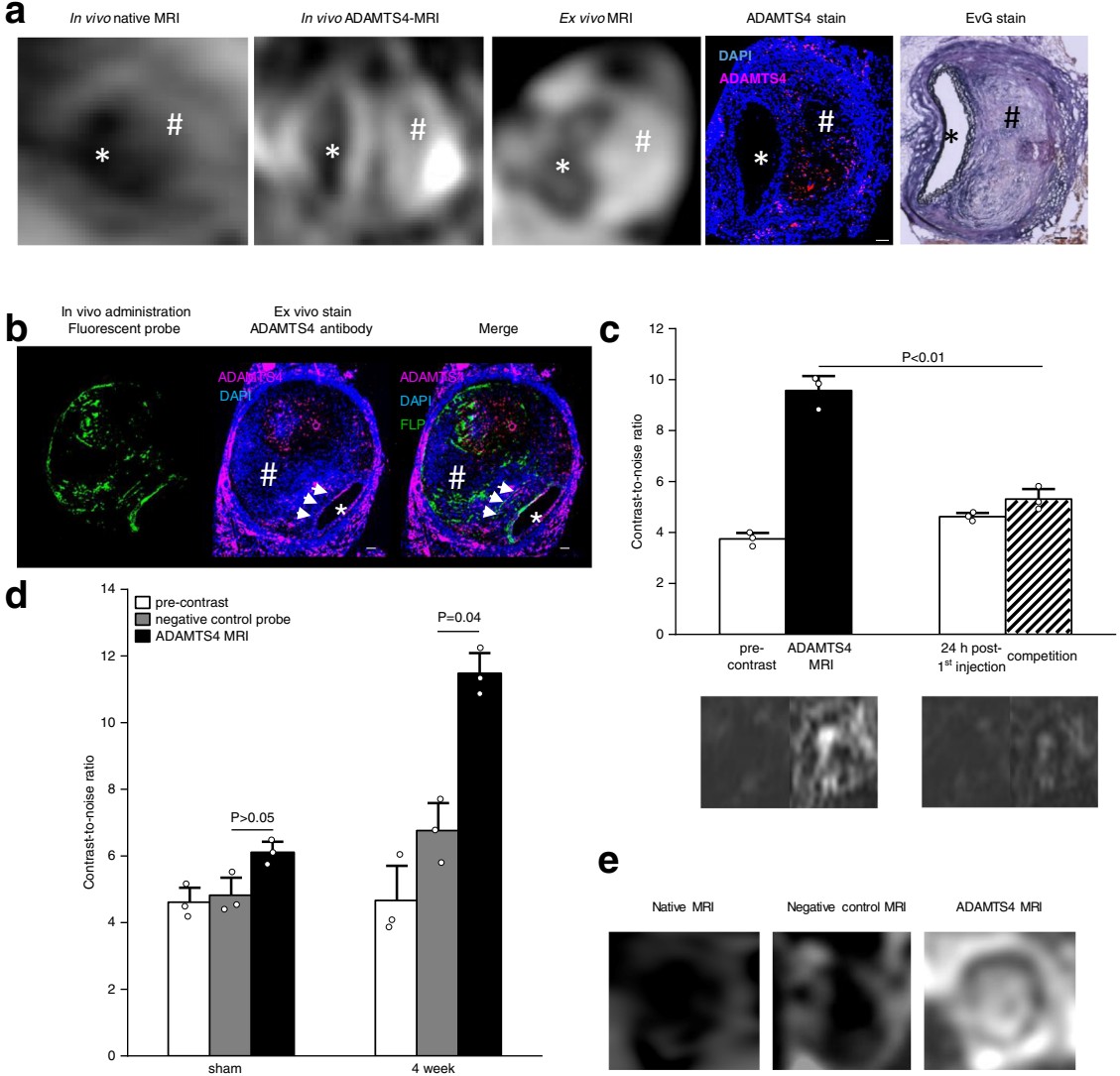

**Fig. 5 Confirmation of the specific binding of the ADAMTS4-specific probe. a** From left to right: In vivo MRI before and after administration of the ADAMTS4-specific probe in one male mouse 4 weeks following aneurysms induction was highly comparable to corresponding ex vivo MRI scan of the abdominal aorta and ex vivo immunofluorescence staining (ADAMTS4 stain) and Elastica van Gieson (EvG) staining. **b** Intravenous administration of the fluorescent variant of the ADAMTS4-specific probe resulted in a strong fluorescence of the aneurysmal wall that was clearly co-localized with areas of ADAMTS4 expression using immunofluorescence staining (white arrows). **c** The competition experiment ($n = 3$ mice) showed a significant decrease in the contrast-to-noise-ratio compared to the administration of the gadolinium-labeled ADAMTS4-specific probe alone, confirming the specific binding of the probe ($P = 0.0025$). Data are presented as mean values ± SEM. **d, e** Using the negative control probe Cys*-Phe-His-Pro-Tyr-Cys*-linker (DOTA-Gd) with a similar size to the ADAMTS4-specific probe, no significant increase in contrast-to-noise ratio (CNR) was measured in vivo in the aortic wall of ApoE$^{-/-}$ mice ($n = 6$). The application of the ADAMTS4-specific probe resulted in a strong and significant increase in CNR in the vessel wall as well in the thrombus. *$P = 0.08$; **$P = 0.04$; Data are presented as mean values ± SEM. FLP Fluorescein-labeled probe, Scale bars indicate 100 μm. * indicate the aortic lumen; # indicates the thrombus area. Source data for **c, d** are provided as a Source Data file. For data analysis, a two-sided, unpaired t-test was used.

($n = 6$). The following sequence for the peptide cycle was used: Cys*-Phe-His-Pro-Tyr-Cys*-linker. A weak signal for this peptide was found in the screening but did not show binding in the SPR-preanalysis as well as in the MST measurement. MR imaging after administration of the negative control probe showed a significantly lower increase in MR signal compared to MR signal enhancement after the administration of the ADAMTS4-specific probe (Fig. 5d, e).

**Ex vivo expression of ADAMTS4 in the aortic wall.** In order to evaluate the expression of ADAMTS4 within the aortic wall, immunohistochemistry was performed. Immunofluorescence

staining of histological sections showed a significant increase in ADAMTS4 expression in mice infused with AngII for 2 and 4 weeks, whereas ADAMTS4 was barely detectable in control mice (Fig. 6a). Immunohistological ex vivo measurements were in good correlation with in vivo T1-weighted MR measurements after administration of the ADAMTS4-specific probe ($y = 0.95x + 7.73$, $R^2 = 0.75$, $p < 0.01$, Fig. 6b). In addition, to evaluate the ADAMTS4 expression in macrophages, the presence of CD68 + macrophages in the aorta was examined. Similar to the ADAMTS4 expression, CD68 + macrophages were most abundant in mice after 4 weeks of AngII-infusion (Fig. 6c). There was a clear colocalization of areas of ADAMT4 expression and CD68 + macrophages as well as a strong correlation between

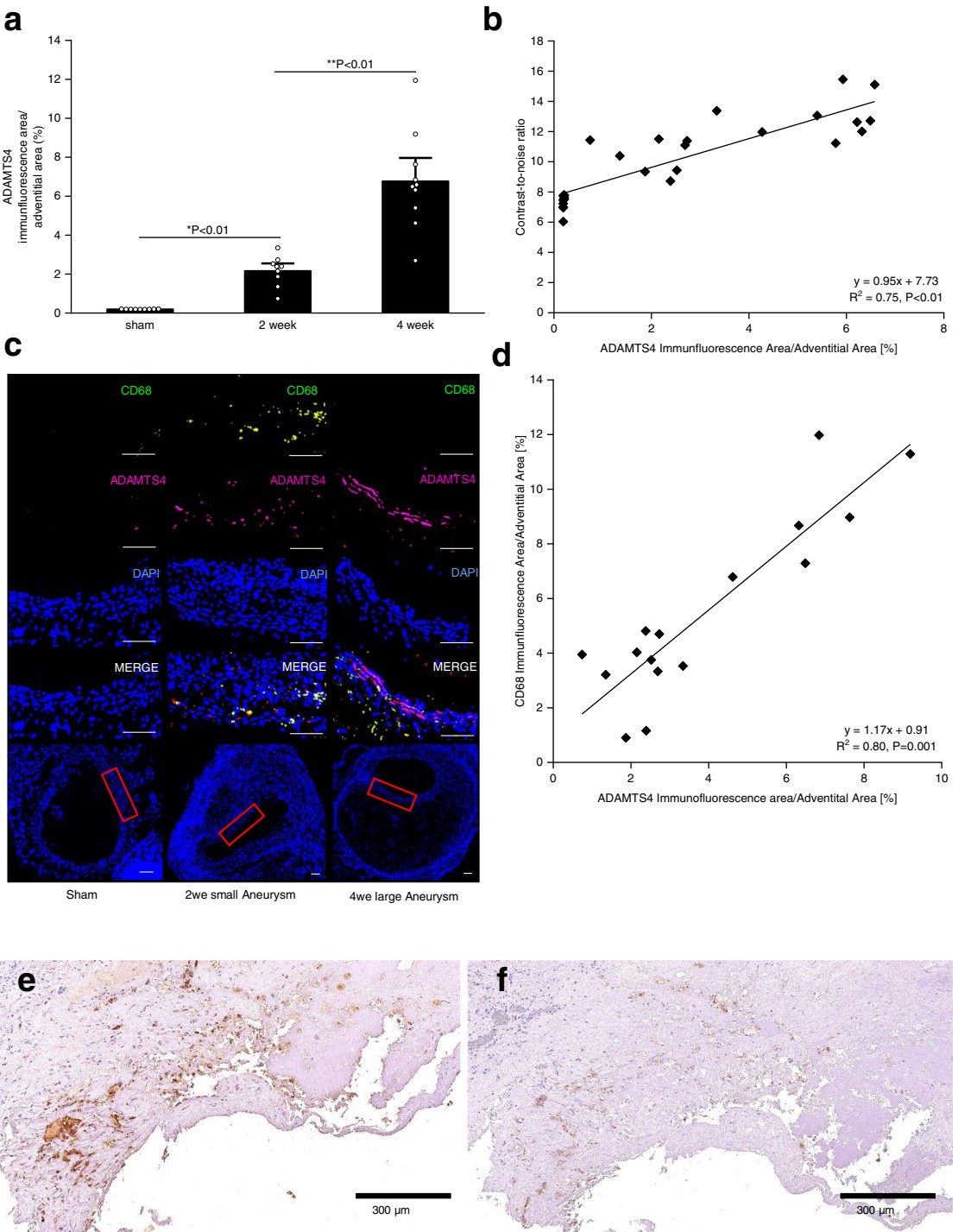

**Fig. 6 Ex vivo expression of ADAMTS4. a** Immunofluorescence analysis showed a significant increase in ADAMTS4-expression 2 and 4 weeks after aneurysm induction ($n = 10$ per group) (*$P = 9.35E-07$; **$P = 6.31E-05$). Data are presented as mean values ± SEM. **b** In vivo contrast-to-noise-ratio (CNR) measurements were in good correlation with ex vivo measurements of ADAMTS4 expression using immunofluorescence ($n = 28$; $y = 0.95x + 7.73$; $R^2 = 0.75$, $P = 4.08E-15$). **c** Immunofluorescence staining showed a strong expression of ADAMTS4 and the associated CD68 macrophages within the aneurysmal wall after two and 4 weeks as well as a clear colocalization between areas of macrophage accumulation and ADAMTS4 expression. Sham-operated mice showed no relevant expression of CD68 or ADAMTS4. Scale bars represent 100 μm. **d** CD68- and ADAMTS4-expression within the aortic wall correlated well, using immunofluorescence analysis ($n = 16$; $y = 1.17x + 0.90$; $R^2 = 0.80$, $P = 0.001$). **e, f** Example of a ruptured human aortic aneurysm, a colocalization of the ADAMTS4 (**e**) with macrophages (**f**) can be appreciated. Sections were incubated with polyclonal ADAMTS4 antibody (Rabbit anti-mouse ADAMTS4) and monoclonal CD68 antibody (Rat anti-mouse CD68). Source data for **a**, **b**, **d** are provided as a Source Data file. For data analysis, a two-sided, unpaired t-test was used.

areas positive for CD68 and ADAMTS4 in relation to the overall adventitial area ($y = 1.17x + 0.09$, $R^2 = 0.80$, Fig. 6c, d).

It has been demonstrated that in a murine model of AAA, ADAMTS4 is expressed by smooth muscle cells in the aortic wall[33]. In addition, ADAMTS4 can also be expressed by endothelial cells and macrophages[22]. To the best of our knowledge, the ApoE$^{-/-}$ + AngII murine model develops fusiform, infrarenal aortic aneurysms, usually accompanied by the formation of an intramural thrombus. In order to evaluate the localization of ADAMTS4, SMCs, and CD68 + cells within the aneurysm (Supplementary Fig. 11), immunofluorescence analysis of 10 μm thick AAA cryosections after 4 weeks of AngII-infusion was performed. We demonstrated a strong colocalization of ADAMTS4 with both SMCs and CD68-positive cells (Supplementary Fig. 10). The incomplete overlap is highly likely due to the fact that not all SMCs and macrophages in the aneurysm express ADAMTS4 simultaneously.

Additionally, we examined human aneurysm samples by histological staining of the sample (77-year-old male patient; ruptured juxtarenal AAA; 115 mm). The staining of histological sections showed an enhanced ADAMTS4 expression in the aortic wall, which was co-localized to the CD68 staining (Fig. 6e, f).

**Gadolinium concentration analysis by using inductively coupled plasma mass spectrometry (ICP-MS).** For the quantification of the gadolinium amount in the aortic wall, ICP-MS was performed in seven mice ($n = 7$). An increase in gadolinium concentration could be observed in AngII-infused mice compared to control mice. Ex vivo measured gadolinium concentrations by ICP-MS were in strong correlation with in vivo CNR measurements following administration of the ADAMTS4-specific probe (Fig. 7a).

**Spatial localization of the gadolinium-based ADAMTS4-specific probe by using laser ablation inductively coupled plasma mass spectrometry (LA-ICP-MS).** To visualize and evaluate the spatial distribution of the gadolinium-containing ADAMTS4-specific probe within the aortic wall, LA-ICP-MS was performed ($n = 3$ per time point, $n = 3$ control mice). A strong colocalization of gadolinium with areas positive for ADAMTS4 was demonstrated (Fig. 7b).

## Discussion

In this study, we identified a low-molecular-weight cyclic highly selective peptide targeted against ADAMTS4 based on an improved OBOC method. Surprising for such a small peptide was the high specificity against ADAMTS4. To clarify the specificity of our probe, we performed MST measurements against biosimilar proteins, including ADAMTS1 and ADAMTS5, which demonstrated lower binding constants. We used also an in vivo competition experiment with a "cold" probe to check for unspecific binding, which was not shown. The probe was subsequently modified for molecular MRI and it is in vivo potential as an MR probe was demonstrated in an AAA mouse model. By using the ADAMTS4-specific probe, a significant increase of in vivo MR enhancement was observed in the wall of aortic aneurysms. Based on the derived strengths of the signal, the molecular probe enabled the prediction of further aneurysm expansion.

Unlike currently used clinical methods, which are based on aneurysmal cross-sectional area or diameter measurements for risk evaluation, ADAMTS4-MRI enables the specific assessment of alterations within the arterial wall at a molecular level.

Peptides are promising candidates for specific molecular probes. They can undergo highly specific and strong binding with different targets, including proteins[34] or sugars[35]. Especially small peptides are interesting candidates for contrast agents due to their favorable distribution in the blood system, their fast blood clearance through the renal system, and their easy synthesis[36]. For obtaining specific peptide binders, several different screening systems were established, whereas phage display represents the currently most widely used method. With phage display, large peptide libraries of nearly every peptide length can be screened[37]. Nevertheless, phage display is limited to proteinogenic amino acids, may be vulnerable to false-positive results by binding to other components as the target of the screening system[38] or enrichment of parasitic sequences, which are amplified by fast-growing bacteria[39]. Overcoming these drawbacks can be challenging and time-consuming.

Among several different combinatorial screening methods, OBOC libraries are particularly well suited for the screening of small cyclic peptide binders[15–17]. Since they are synthetic, they are easier to control compared to biological methods and parasitic

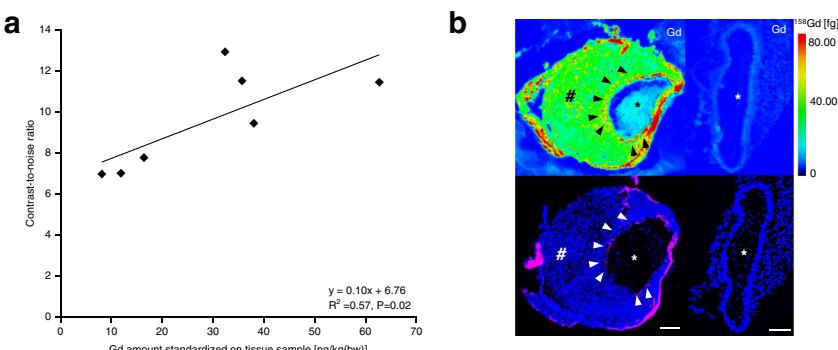

**Fig. 7 Correlation of in vivo MRI with inductively coupled plasma mass spectrometry (ICP-MS) and spatial localization of Gd using laser-ablation inductively coupled plasma mass spectrometry (LA-ICP-MS). a** Inductively coupled plasma mass spectrometry (ICP-MS) analysis showed a good correlation between contrast-to-noise-ratio measurements and Gd amount in the aortic wall ($n = 7$; $y = 0.10x + 6.77$; $R^2 = 0.57$, $P = 0.02$). **b** First row: Laser-ablation inductively coupled plasma mass spectrometry visualized the gadolinium-based ADAMTS4-specific probe within the aortic wall (black arrowheads) of a mouse 4 weeks after aneurysm induction (left hand) and of a sham-operated mouse (right hand) by the Gd-signal. Second row: A clear colocalization of gadolinium-accumulation with areas positive for ADAMTS4 using immunofluorescence was observed (white arrowheads). Scale bars indicate 100 μm. * indicate the aortic lumen; # indicates the thrombus area. Source data for a are provided as a Source Data file. For data analysis, a two-sided, unpaired $t$-test was used.

sequences can be avoided. Nevertheless, positive hits are manually picked for analysis, which can lead to false-positive results or loss of positive beads. Automated systems often require additional instrumentation[40] or complex consumables[41]. Recently, an easy and cost-efficient combined approach was introduced, which enables screening and peptide sequencing on the same chip to bypass the manual separation[18]. With this approach, linear peptide binders, with micromolar affinities could be detected. To identify peptides with higher binding affinities, we expanded this technique to cyclic peptides. Sequencing of cyclic peptides can be difficult[42] and often requires special reagents, like photochemical linkers[43] or special protection group strategy, to form cleavable cycles[44,45]. In our study, we used a disulfide for ring closure. To avoid ring-opening reactions, we applied the ladder sequencing of St. Hilaire et al. to our system[27]. Therefore, the ring remained closed, while the ladder peptides stayed linear. In the next step, the sequence could be read out by MALDI-TOF-MS. Only the thiol group of the first cysteine in the ladder peptides needs to be alkylated. A further advantage of the combined on-chip approach is the regeneration process during the screening. Thereby, false-positive signals can be eliminated since a prescan provides information about the binding against the detection antibodies or the autofluorescence of the beads[18,45]. Furthermore, we could increase the selectivity by screening against the structurally similar ADAMTS5 (Supplementary Fig. 1a). This selectivity could be confirmed by our MST measurements (Supplementary Fig. 3a, b). While we measured a nanomolar affinity of our chosen candidate against ADAMTS4, we could not detect binding against ADAMTS1 and ADAMTS5 in this concentration range. Additionally, we found a potential binding cleft in ADAMTS4 (Fig. 2c, d), which could enable a strong interaction with the ADAMTS4-specific peptide. Especially, the arginine in the peptide, as well as the glutamic acids in the binding cleft, apparently has a strong influence on the binding, since, by exchange of one of them, the binding disappears (Supplementary Fig. 3a, c and Supplementary Fig. 6a, b). The glutamic acids in the cleft are not found in ADAMTS1 or ADAMTS5. Our negative control peptide, which showed only a small increase of fluorescence in our screening against ADAMTS4, showed no affinity against ADAMTS4. This provides evidence, that only a high increase in the fluorescence in our on-chip screening led to affine peptides with high selectivity. The exchange of one of the arginines in the peptide sequence leads to a loss of binding, which is consistent with our computational mutation study. In this study, the mutation of the glutamic residues in our proposed binding cleft with glutamine leads also to a loss of binding. Although we used only a small library of ~22,000 beads, we were able to identify small peptide binders with high selectivity against ADAMTS4 with nanomolar affinities. Furthermore, the probe shows superior characteristics like high solubility, degradational stability, and low toxicity, which are crucial parameters for a later translation into human medicine.

The pathogenesis of AAAs mainly relied on tissue samples obtained during surgical interventions at end-stage of AAAs. At this late stage, pathologic features of the aortic wall include the expression of elastic fibers, adventitial hypertrophy, and accumulation of pro-inflammatory cells[46,47]. Very limited data is available on the composition of early-stage human AAAs. An increasing amount of data is, however indicating, that aortic dissection could be the initiating event for the development AAAs in humans[48,49].

There is also accumulating evidence that the extracellular matrix enzyme ADAMTS4 is strongly upregulated in an aneurysm, and therefore represents a promising in vivo target to assess the risk of rupture[19–21]. Its expression is specifically linked to fatal aortic rupture and unstable plaque formation[22]. Using the ADAMTS4-specific probe, a significant increase of in vivo MR

signal was observed only 2 weeks after AAA induction, while with a peptide of similar size as a negative control, we could detect only a small increase, which may be caused by a non-specific absorption of the probe into the ECM. Four weeks after AAA induction, the binding and MR signal in the aortic wall (enhancement) after the administration of the ADAMTS4-specific probe further increased (Fig. 3a–d).

We[50,51] and others[52,53] have shown the potential of molecular imaging not only for the diagnosis of AAAs but also for the prediction of further aneurysm development or even rupture. By performing a longitudinal study, it was shown that the molecular probe has the potential to predict further aneurysm expansion (Fig. 3d, e). Animals with significantly higher MR signal enhancement in week 1 after AAA induction showed an extensive AAA progression from week 1 to week 4 compared to mice with only a slight increase in AAA diameter during the same period. These findings suggest that a high ADAMTS4 expression within the aortic wall may promote a more rapid AAA progression.

In the clinical setting, there is a common consensus that patients with AAAs >55 mm should undergo surgery[6]. There is, however, still controversy surrounding the management of asymptomatic medium-sized AAAs (40–55 mm)[6,8]. No biomarker is currently available, which could help to better characterize aortic aneurysms in patients and to predict their risk of rupture[6,8]. Additionally to the longitudinal study, we performed also an early detection study, in which we could show, that our probe has also the potential as a rupture prediction marker. After 4 days of AngII-infusion, we could separate the animals by ADAMTS4-MRI levels into three different groups. Animals with the highest MRI signal suffer a fatal AAA rupture, while animals with moderate contrast enhancement developed AAA but survived for 4 weeks. Animals, which did not develop AAA, did not show a significant increase in ADAMTS4-MRI level ($P > 0.05$) (Fig. 4a-d).

Additionally, we detected an increasing accumulation of macrophages within the aneurysmal wall, which were co-localized with areas of ADAMTS4 expression (Fig. 6c, d). The same colocalization was observed in ruptured AAA in humans (Fig. 6e, f). Since inflammation was shown to be a key process in AAA development, inflammatory activity represents a promising in vivo biomarker for the characterization of aneurysm development and progression. The group of ADAMTS matrix metalloproteinases was shown to contribute to the development of several inflammatory diseases, like arthritis and vascular diseases[54,55]. ADAMTS4 and ADAMTS1 may be involved in atherosclerotic plaque formation and stability, as both have been found to be highly expressed in macrophage-rich areas of human atherosclerosis plaques[21,56]. Especially ADAMTS4 levels have been shown to be increased in aortic tissue from patients with thoracic aortic aneurysms[54,57,58]. An increased expression of ADAMTS4 may promote aneurysm progression by degrading ECM components and facilitating inflammatory cell infiltration[57]. In support of its clinical relevance, Ren et al. have recently shown that ADAMTS4 contributes to the formation of aneurysms dissection and rupture in both thoracic and abdominal aortic segments[54].

Ultrasonography, computed tomography angiography (CTA), and magnetic resonance angiography (MRA) are the most commonly used techniques for screening, initial assessment, and follow-up of patients with AAAs[59]. For small asymptomatic AAAs (<35 mm), close follow-up is recommended, while for large AAAs (>55 mm) surgery or endovascular repair is the recognized management[6]. In clinical practice, the diameter of AAAs is still the only recognized feature to predict the risk of rupture. While there is general agreement that large aneurysms should undergo definite treatment, there is still controversy regarding the management of medium-sized aneurysms (40–55 mm). This

controversy reflects the limitation of diameter as a marker of risk of rupture. For the more detailed assessment of patients, a novel marker is therefore urgently needed. ADAMTS4-MRI could improve the in vivo characterization of aortic aneurysms. As upregulation of ADAMTS4 was shown to be associated with an increased risk of expansion with potential following rupture of AAAs, ADAMTS4 could provide important information for more accurate risk stratification of this patient group. Peng et al. developed a method for ex vivo testing of ADAMTS4, which shows high sensitivity[60]. While this approach shows great potential for ex vivo ADAMTS4 detection, the use in vivo as an MR imaging agent might be difficult since after cleaving the peptide, both fragments have no further binding to ADAMTS4. However, for MRI the period of time, in which the contrast agent binds to its target is important to reduce the background[12,13]. Comparing the ADAMTS4-MRI signal enhancement with the aortic area in a receiver operating characteristic (ROC), we could demonstrate, that the ADAMTS4-MRI level is a stronger predictor for a fatal AAA rupture with an area under the curve of 0.96 (ADAMTS4-MRI level) to 0.78 (aortic area) (Fig. 4e, f).

There is great potential in translating the use of the ADAMTS4-specific probe into a clinical setting. The molecular composition and size of the molecular probe is comparable to agents already used in clinical practice[61] making adverse effects substantially less likely compared to larger molecules, e.g., antibodies and nanoparticles[62]. The screening was performed by using recombinant human ADAMTS4, which enables the application of the found peptide for human imaging. The probe shows no relevant toxicity against human cells and is caused by the use of only canonical amino acids and DOTA as a chelating agent; side reactions are decreased to a minimum. Furthermore, the probe shows superior characteristics like high solubility and degradational stability, which are crucial parameters for translation. Further advantages of this study regarding clinical translation include the administration of the imaging agent at a clinical dose. Relaxation, rotational correlation, and signal properties can therefore be directly translated to human application. In addition, several properties of our ADAMTS4-specific probe are highly comparable to contrast agents already used in clinical practice, including molecular size and composition. Furthermore, the clearance of the ADAMTS4 appropriate is comparable[61] to approved gadolinium-based MR contrast agents allowing for early imaging after injection with minimal background signal and thus maximal target to blood contrast to noise.

This study has certain limitations. There are differences between human AAAs and those aneurysms formed in the used mouse model. In AngII-infused ApoE$^{-/-}$ mice, AAAs develop in the suprarenal region[63], whereas human AAAs develop in the infrarenal region. AAAs in human diseases are generally not caused by dissection or intramural hematoma. Even though aortic dissections often lead to the development of aneurysms, the vast majority of AAAs have a different pathophysiology. Although the ADAMTS4-specific probe showed a high affinity against ADAMTS4 and good colocalization with ex vivo immunofluorescence, we cannot fully exclude cross-reactivity with other proteins not screened in this study.

In summary, we identified a low-molecular-weight cyclic peptide with high selectivity against ADAMTS4 and favorable properties for an in vivo MR application, using an improved OBOC method. The probe was successfully modified for molecular MRI, and its in vivo potential as an MR probe was demonstrated in an AAA mouse model. Beyond aneurysms, an ADAMTS4-specific probe may also be useful for the early detection and the in vivo characterization of cardiovascular diseases, including atherosclerosis, as ADAMTS4 also plays a key role in these diseases.

## Methods

### Ethical statement
All procedures and animal maintenance were performed according to the guidelines and regulations of the Federation of Laboratory Animal Science Associations (FELASA) and the local Guidelines and Provisions for Implementation of the Animal Welfare Act. All animal experiments were approved by the local authorities (Lageso Berlin). The use of human samples was approved by the ethics committee of the TU Munich.

### Materials and chemicals
The solvent dimethylformamide (DMF, 1.00387) was purchased from Merck (Darmstadt, Germany), ethanol 99.8% for analysis, absolute (493511), from Sigma Aldrich (St. Louis, MI, USA), and diethyl ether (2362) and dichloromethane (2356) from Th. Geyer GmbH & Co. KG (Renningen, Germany). Lab water was obtained from a Milli-Q water purification system (Millipore, Bedford, MA, USA) with a resistivity of >18.2 Ω and TOC-value of <5 ppb. The resin for peptide synthesis TentaGel HL NH$_2$ (HL12902) was purchased from Rapp Polymere GmbH (Tübingen, Germany). The linker 4-hydroxymethylbenzoic acid (HMBA, AB131645) was obtained from abcr (Karlsruhe, Germany). 4-methyl morpholine (NMM, M56557) was purchased from Sigma Aldrich.

The protected amino acids Fmoc-Gly-OH, Fmoc-ProOH, and Fmoc-Phe-OH were purchased from Bachem (Bubendorf, Switzerland); Fmoc-Trp(Boc)-OH, Fmoc-Arg(Pbf)-OH, Boc-Arg-OH, Boc-Glu(OtBu)-OH, and Boc-Ser-OH from J&K Chemical Ltd. (Shanghai, China); Fmoc-Tyr(tBu)-OH, Fmoc-His(Trt)-OH, Fmoc-Glu(OtBu)-OH, Boc- Phe-OH, Boc-Pro-OH, Boc-Trp-OH and Boc-Tyr-OH from Sigma Aldrich; Fmoc- Ser(tBu)-OH, Fmoc-Lys(Boc)-OH, Fmoc-Cys(Trt)-OH, Fmoc-Asn-OH, Boc-Asn-OH, Boc-Val-OH, Boc-Gly-OH, and Boc-His(Tos)-OH from IRIS Biotech GmbH (Marktredwitz, Germany). Fmoc-TTDS-OH (FAA1568.0005) was purchased from IRIS Biotech GmbH. The coupling reagent O-(1H-6- Chlorobenzotriazole-1-yl)-1,1,3,3-tetramethyluronium hexafluorophosphate (HCTU, 2112.3) and diisopropyl carbodiimide (DIC, 6981.2) were obtained from Carl Roth GmbH & Co. KG (Karlsruhe, Germany). For deprotection, triisopropylsilane (TIS, 233781) was purchased from Sigma Aldrich, and trifluoroacetic acid (TFA, SOL-011.0500) was purchased from IRIS Biotech GmbH. ADAMTS4 (30 400 102), ADAMTS1 (30 400 402), and ADAMTS5 (30 400 502) for screening, SPR, and MST were purchased by BioTeZ Berlin-Buch GmbH (Berlin, Germany).

The monoclonal anti-His Tag antibody produced in mouse, 1 mg/mL, clone 6G2AG, Protein A purified, lyophilized (ABIN387699) was purchased from antibodies-online GmbH (Aachen, Germany), anti-Mouse IgG-Atto 633 antibody produced in goat (78102- 6 1ML-F), and Bovine Serum Albumin (A7906) were purchased from Sigma Aldrich. The chemicals Tween-20 (P7949) HEPES (H4034-100G), NaCl (71376), EDC (E1769), ethanolamine (02400), guanidine hydrochloride (G3272-25G), ammonia water (28–30%, 221228) were obtained from Sigma Aldrich. PBS buffer (1X, Dulbecco's) (A0964) and EDTA (131669) were purchased from AppliChem GmbH (Darmstadt, Germany). The MALDI matrix 2,5-dihydroxyacetophenone (DHAP, A12185) was purchased from Alfa Aesar (Ward Hill, USA) from Sigma.

C57BL/6 mouse primary aortic endothelial cells were purchased at Cellbiologics (Chicago, Illinois, USA) and human aortic endothelial cells were purchased by ATCC® (Manassas, Virginia, USA). Cell medium supplement kit (0.5 mL VEGF, 0.5 mL ECGS, 0.5 mL heparin, 0.5 mL EGF, 0.5 mL hydrocortisone, 5.0 mL L-glutamine, 5.0 mL antibiotic-antimycotic solution and 25.0 mL FBS) was purchased from Cellbiologics (Chicago, Illinois, USA), while MTT (3-(4,5-dimethylthiazol-2-yl)-2,5- diphenyltetrazolium bromide) reagent was purchased from Abcam (Cambridge, United Kingdom). The CellTiter-Glo and CytoTox-One assay were purchased from Promega (Madison, Wisconsin, USA).

Angiotensin (angio II) was purchased from Merck (Darmstadt, Germany), Medetomidine (Cepetor) was purchased from cp-pharma (Burgdorf, Germany), Fentanyl and Midazolam were purchased from Rotexmedica (Trittau, Germany; acquired by Panpharma S.A., Luitré, France), Atipamezole (Antisedan) was purchased from Orion Corporation (Espoo, Finland), Naloxone was purchased from B. Braun Melsungen AG (Melsungen, Germany) and Flumazenil was purchased from Hameln pharma plus GmbH (Hameln, Germany). Staining solutions for EvG and HE were purchased from Morphisto Ltd. (Frankfurt am Main, Germany). ROTI®Mount FluorCare DAPI (HP20.1) was purchased from Carl Roth GmbH & Co. KG (Karlsruhe, Germany). Dako REAL Antibody Diluent was purchased from Dako (Denmark, acquired by Agilent, California, USA). The polyclonal anti-mouse ADAMTS4 antibody produced in rabbits (Invitrogen) was purchased from Thermo Fisher Scientific (Waltham, Massachusetts, USA), monoclonal anti-mouse CD68 antibody produced in rat (clone FA-11) was purchased from Bio-Rad (California, USA), polyclonal secondary anti-rabbit antibody Alexa Fluor 647 produced in donkey (Donkey anti-Rabbit IgG, Thermo Fisher Scientific, Germany) and anti-rat Alexa Fluor 568, produced in goat were purchased from Thermo Fisher Scientific. The anti-smooth muscle actin, the rabbit-anti-mouse primary antibody was purchased from Elabscience (E-AB-34268.20, Elabscience, TX, USA). The ADAMTS4-specific peptide and the negative control peptide with a DOTA label and the ADAMTS4-specific peptide with a FAM-5 label were purchased from peptides&elephants GmbH (Hennigsdorf, Germany) with >95% purity each. ADAMTS4-specific peptide, the negative control peptide, and the alanine scan peptides with a Cy5 label were purchased from Peptide Specialty Laboratories GmbH (Heidelberg, Germany) with >95% purity

each. GdCl$_3$·6 H$_2$O (99.999% trace metals basis, 203289-5 G), EuCl$_3$·6 H$_2$O (99.99% trace metals basis, 203254-1 G), and ammonium acetate (73594-25G-F) were purchased from Sigma Aldrich.

Nitric acid (68%, puriss.) and hydrogen peroxide (30%, p.a.) were purchased from Carl Roth GmbH & Co. KG (Karlsruhe, Germany). Gadolinium ICP Standard CertiPUR was purchased from Merck KGaA (Darmstadt, Germany).

**Synthesis of HMBA linker and spacer on resin.** The HMBA linker was attached to 1.006 g TentaGel beads (HL12902: TentaGel® HL NH$_2$; particle size: 75 µm; capacity: 0.4–0.6 mmol/g; ca. 476 µmol NH$_2$) as described in the literature[18]. Therefore, the beads were swollen for 45 min in 15 ml DMF. The DMF was drained and HMBA (380 mg, 1166 µmol, 2.5 eq), HCTU (980 mg, 1166 µmol, 2.5 eq), and NMM (520 µl, 2332 µmol, 5 eq) were preactivated for 2 min in 12 ml DMF. The solution was added to the resin and mixed for 2 h. The solution was drained and the synthesis was repeated three times. The first amino acid of the spacer, glycine, was coupled with the anhydride method. Therefore, 1.4 g of Fmoc-glycine (4.7 mmol, 10 eq) were suspended in 4 ml DCM. DMF was added until the Fmoc-glycine was completely dissolved. The solution was cooled down to 0 °C and 360 µl of DIC (2.35 mmol, 5 eq) were dissolved in 800 µl DCM and added to the solution. The solution was stirred for 20 min in an ice bath until a white solid is formed. 952 µl of a 50 mM stock solution of DMAP in DMF (47 µmol, 0.1 eq) were added until the solid is completely dissolved and added to the resin for 1 h. The solution was drained and the resins were washed two times with DMF and three times with DCM. The step was repeated three times. The Fmoc cleavage was performed with 10 ml of 20% piperidine in DMF for 15 min. The solvent was removed, and the Fmoc cleavage was monitored by UV/Vis spectroscopy at 300 nm. The resin was washed three times with DMF. Afterward, the next amino acids were coupled by standard procedure. For each coupling step, four equivalents of the Fmoc-protected amino acids (1880 µmol), four equivalents of HCTU (778 mg, 1880 µmol), and eight equivalents of NMM (414 µl, 3760 µmol) in 10 ml DMF were used, and the coupling was repeated twice. For the TTDS-coupling, five equivalents of Fmoc-TTDS (1275.2 mg, 2350 µmol) and HCTU (972.2 mg, 2350 µmol) and 10 equivalents of NMM (517 µl, 4700 µmol) were used. After each amino acid elongation, the Fmoc protecting group was removed with 10 ml of 20% piperidine in DMF for 15 min. The solvent was removed, and the Fmoc cleavage was monitored by UV/Vis spectroscopy. The spacer sequence was Ser-TTDS-TTDS-Ser-Arg-Gly. For a description of the exact amount of used material please refer to Supplementary Table 1.

**Synthesis of library.** 300 mg of the previously synthesized resin with peptide linker (loading 476 µmol/g) were used for the library. First, Fmoc-cysteine was coupled to the resin by using the same amounts as previously described. Therefore, Fmoc-L-Cys(Trt)-OH (330 mg, 564 µmol, 4 eq) were preactivated with HCTU (233 mg, 564 µmol, 4 eq) and NMM (125 µl, 1128 µmol, 8 eq) in 4 ml DMF for 2 min and added for 1 h to the resin. The solvent was drained and the beads were washed once with DMF. The coupling was repeated and the resin was washed three times with DMF. For the Fmoc removal, 3.5 ml of 20% piperidine in DMF was added to the resin for 15 min. After Fmoc removal, the resin was washed several times with DCM, and by suspending in DCM, the beads were split in 11 different synthesis reactors. The beads were washed several times with DMF, and in each reactor, one Fmoc-aa (amino acid) combined with 7% of the related Boc-aa was coupled twice per round. For a description of the exact amount of reagents please refer to Supplementary Table 2. Each coupling step was performed for 1 h. Afterward, the resin was washed with DMF, and all beads were combined in one big reactor. Fmoc was cleaved like previously described with 3.5 ml 20% piperidine in DMF, and the beads were split again. After four couplings for the library, Fmoc-L-cysteine was coupled as the last amino acid to the whole resin. Therefore, the same procedure was used, as for the first cysteine. The resin was washed with DMF twice and afterward with ultrapure water three times. The last step was the cleavage of the permanent protecting groups with a mixture of 5 ml of 95% TFA, 2.5% ultrapure water, and 2.5% TIPS three times for 2 h each. The resin was washed with pure TFA, DMF, and ultrapure water. The cyclization was performed overnight in a mixture of 50% DMSO in PBS. The beads were washed three times with ultrapure water. Afterward, 2 ml of a mixture of 0.5 M iodoacetamide in 70% MeOH and 30% PBS were added to the resin for 30 min. After removing the mixture, the beads were washed three times each with ultrapure water, DMF, DCM, and ether and dried under vacuum.

**Screening against ADAMTS4.** A glass slide was laminated with double-sided tape with electrical conductivity. By using a sieve with 100 µm$^2$ pore size, the beads were immobilized on one slide. The slide contains around 40.000 beads in a well-ordered way. The slide was soaked overnight in PBST (PBS with 0.1% Tween-20) on a shaking platform at 160 rpm. In the next step, the chip was incubated with 2 µg of monoclonal anti-His-antibody (produced in mouse), 1 mg/mL, clone 6G2AG, Protein A purified, lyophilized (ABIN387699) antibodies-online GmbH (Aachen, Germany) in 10 ml PBST with 1% BSA under shaking at 160 rpm for 1 h. The chip was washed two times in PBS, and afterward, the chip was incubated with 2 µg of polyclonal anti-mouse antibody (gG-Atto 633 antibody produced in goat (78102-6 1ML-F) Sigma Aldrich); polyclonal anti-mouse ADAMTS4 antibody produced in

rabbit (ADAMTS4 Polyclonal Antibody, PA1-1749A, Invitrogen), Thermo Fisher Scientific (Waltham, Massachusetts, USA) in 10 ml PBST under shaking with 160 rpm for 1 h. The chip was then washed twice in PBS and once in ultrapure water. Afterward, the chip was dried under vacuum for 20 min and scanned in a microarray scanner MArS (Ditabis AG) with a wavelength of 635 nm. This prescan is necessary since both antibodies could also work as a target, and thus, all beads, which demonstrate binding against the antibodies, were identified. Based on this approach, the number of false-positive hits in the subsequent screenings was reduced and positive hits from the prescan were excluded in the following screenings. Additionally, the non-specific binding of the antibodies to the beads was characterized. This non-specific binding led to a small increase of the fluorescence, which was eliminated from the main screening. Subsequently, the chip was soaked again overnight in PBST on a shaking platform at 160 rpm, followed by a 40 min denaturation in 6 M guanidinium chloride. Following this, the chip was washed twice in 0.1% TFA in ultrapure water, ultrapure water, and PBS each. In the next step, the chip was soaked for 4 h in PBST. Consequently, the chip was incubated with a solution of 2 µg ADAMTS4 in 10 ml PBST with 1% BSA on a shaking platform at 160 rpm overnight. The chip was washed twice with PBS, and both antibodies were applied in the same way as previously described. The chip was washed as described and scanned again. Then the chip was recovered with guanidium chloride, and the same protocol was repeated with ADAMTS5. After recovery, the chip was washed with 0.1% TFA twice and with ultrapure water and dried for 20 min under vacuum and scanned again as postscan.

**Evaluation of the scans and MALDI MS examination of the hits.** All four scans were merged in the ImageJ software (1.51, National Institute of Health, USA) and analyzed. The fluorescence intensity was measured for 9 pixels (225 µm$^2$) as the region of interest (ROI) and the mean value was used. The chip coordinates of the hits were used for the spatial resolution in the MALDI-TOF-MS instrument (Bruker autoflex II smartbeam™). The peptides were cleaved from the resin in an ammonia gas atmosphere overnight. Afterward, the 1.5-DHAB matrix (6 ml of 4:1 EtOH: ultrapure water, 30 mg 1.5-DHAB, 10.5 mg ammonium citrate dibasic) was applied to the chip via an airbrush system. Finally, the chip was measured with a MALDI-TOF-MS.

**Resynthesis of the positive hits.** The resynthesis was performed as previously described. 30 mg of the previously synthesized resin with peptide linker were used per peptide. Each coupling step was only performed once for 1 h. Therefore, the amino acid was preactivated with HCTU and NMM in DMF for 2 min (for exact amounts, please refer to Supplementary Table 3). Afterward, the solution was added to the resin. After 1 h, the solution is drained and the resin was washed three times with DMF. Fmoc was cleaved with 1 ml of 20 % piperidine in DMF for 15 min and the resin was washed afterward with five times DMF. The last step was the cleavage of the permanent protecting groups with a mixture of 1 ml of 95% TFA, 2.5% ultrapure water, and 2.5% TIPS two times for 2 h each. The resin was washed with pure TFA and ultrapure water. The cyclization was performed overnight in a mixture of 40% DMSO in PBS. The beads were washed three times each with 20% DMSO in PBS, ultrapure water, and ACN and dried under vacuum. The peptides were cleaved from the resin in an ammonia gas atmosphere for 6 h and extracted two times each 0.5 ml 4:3:3 ACN: ultrapure water:AcOH. The solutions were lyophilized to a white solid (crude yields: Pep7 15 mg (69%), [M + H]$^+$ calculated 1523.76; found 1524.15; Pep11 14 mg (58%), [M + H]$^+$ calculated 1679.81; found 1680.27; Pep12 21 mg (81%), [M + H]$^+$ calculated 1815.83; found 1816.36; Pep15 17 mg (68%), [M + H]$^+$ calculated 1757.83; found 1758.35; Pep16 17 mg (71%), [M + H]$^+$ calculated 1669.85; found 1670.42). For MALDI-ToF-MS data, please refer to Supplementary Fig. 11–15 and Supplementary Table 4. The peptides were used directly for the SPR without further purification.

**SPR analysis of the resynthesized peptides.** Surface Plasmon Resonance (SPR) experiments were performed on a Reichert SR7500DC dual-channel SPR system (Reichert, USA). 0.01 M HEPES pH 7.4, 0.15 M NaCl, 3 mM EDTA, 0.005% Tween-20 was used as flow buffer. Anti-His-Ab was immobilized on a gold biosensor chip HC1500 (XanTec bioanalytics, Germany) using the following amine coupling protocol: 0.2 M EDC and 0.1 M NHS were dissolved in ultrapure water and directly injected over both channels for 5 min (20 µl/min) twice. Afterward, anti-His-Ab (0.45 µg/ml) in acetate buffer 10 mM; pH 5.15 was injected once over both channels, followed by quenching with 1 M ethanolamine (pH 8.2) over both channels. ADAMTS4 dissolved in the running buffer (4.5 µg/ml) was injected for 15 min (5 µl/min) over both channels. Afterward, the chip was flushed with the running buffer until the stable baseline was reached. The difference at this point was 12421.3 µRUI. By calculating the capture amount of ADAMTS4 on the sample channel, we reached around ¼ capturing of ADAMTS4 of the R$_{max}$. All peptides were injected for 3 min association and 7 min dissociation (25 µl/min) with the following concentrations 3.125 µM, 12.5 µM, 50 µM, 200 µM, 800 µM. Before and after the run, a blank was measured. For peptides 11, 12, and 16 the concentrations showed to high signal. Therefore, the chip was eluted with 6 mM phosphate buffer (pH 2.3) and reincubated with ADAMTS4. For peptides 11 and 12 the following concentrations were used: 10 nM, 100 nM, 1 µM, 10 µM, 50 µM and 100 µM, for peptide 16: 1 nM, 10 nM, 100 nM, 1 µM, 10 µM and 50 µM. The calculation was

done with the TraceDrawer (1.6.1, Ridgeview Instruments, Uppsala, Sweden), Scrubber2 (BioLogic Software, Canberra, Australia), and Origin 2022 (OriginLab Corporation, Northampton, MA) by using the steady-state for analysis.

**MST analysis of the Cy5-labeled peptide**. Microscale thermophoresis (MST) was performed on a Monolith Pico (NanoTemper, Munich, Germany). The Cy5-labeled peptides were diluted into the MST buffer (150 mM NaCl, 50 mM TRIS-HCl, 5 mM $CaCl_2$, and Tween-20) to reach a dilution of 6 nM. ADAMTS4, ADAMTS5, and ADAMTS1 were directly used with the supplied concentration. For ADAMTS4, the starting concentration was 4.96 μM, while for ADAMTS1 it was 4.88 μM and for ADAMTS5 it was 2.46 μM. Since we saw a loss of binding after centrifugation, the use of protein concentrators were inappropriate. 20 μl of the protein solutions were transferred into a PCR tube. 10 μl of this first vial was transferred into a second vial and mixed by pipetting with 10 μl of the MST buffer. The dilution series was continued 14 times. From tube 16, 10 μl were removed. Into all 16 tubes, 10 μl of the respective peptide solution were added and mixed by pipetting. The samples were incubated for 20 min and subsequently transferred to premium capillaries (NanoTemper, Munich, Germany). The capillaries were transferred into the Monolith tray and measured with 10% excitation power on the pico-red excitation on high MST power at 25 °C. Analysis was performed with MO.Affinity Analysis v3.0.1 (NanoTemper, Munich, Germany) and Origin 2022 (OriginLab Corporation, Northampton, MA). We detected a ligand-dependent fluorescence change for some of the systems. We checked for ADAMTS4 and the ADAMTS4-specific-Cy5-labeled probe via a saturation test for specifying the ligand-dependent fluorescence change. Therefore, the three highest ligand concentrations and the three lowest ligand concentrations of the measurement were used and centrifuged for 15 min at 16060 x g at 20 °C. Of each concentration, 7 μl were picked and mixed with 7 μl of a 10 M urea solution each. The samples were heated at 95 °C for 5 min and measured in the MST. The fluorescence difference between the concentrations was afterward at the same level. Each measurement was repeated three times independently and the binding constant was calculated by merging all three measurements. The data showed a sigmoid curve and were fitted by the MO.Affinity Analysis v3.0.1. The data were normalized using MO Affinity Analysis v3.0.1 and the fraction bound [−] was shown. For the three measurements (negative control peptide, ADAMTS4-specific peptide-Arg1, ADAMTS4-specific peptide-Arg2) the fraction bound could not be calculated. Therefore, the percentage variance of the values to the average was plotted.

**Protein docking**. The peptide-protein docking program ATTRACT[64,65] was used for predicting putative interaction geometries between a model structure of ADAMTS4 and the ADAMTS4-specific cyclic peptide. A structural model of ADAMTS4 was downloaded from the AlpfaFold2-Server (https://alphafold.ebi.ac.uk/entry/O75173)[31]. The cyclic peptide structure was generated (sequence: Cys*-Ser-Arg-Arg-Gly-Cys*-Ser-Gly) by comparative modeling based on the cyclic peptide structure pdb2MGO with the same number of residues in the cyclic segment between disulfide-linked Cys residues. The top 50 (best scoring) docking solutions were screened for complexes with all residues of the cyclic peptide in contact with the ADAMTS4 protein since the Ala scan suggested that all residues are important for efficient binding. With this criterion, a solution was found that indicated simultaneous contact of the Arg residues in the cyclic peptide with two separate Glu residues (Glu401, Glu402) on ADAMTS4 in a binding cleft on the protein surface. The Glu residues are specific for ADAMTS4 and are not found in ADAMTS1 or ADAMTS5. The docked structure was solvated in TIP3P water and energy minimized (5000 steps) and heated to 310 K during 0.2 ns Molecular Dynamics (MD) simulations employing positional restraints on the protein and peptide and using the Amber18 package[66] The equilibrated structure served as the start structure for unrestrained MD simulations at 310 K (37 °C) and a pressure of 1 bar that indicated no dissociation on the simulation time scale of 0.1 μs. As a further control, a variant of the docked human cyclic peptide-ADAMTS4 complex with the Glu401 and Glu402 replaced by Gln residues (Glu401Gln, Glu402Gln) was generated. In addition, a docking model was generated for murine ADAMTS4 based on the AlpfaFold2-Server entry: https://alphafold.ebi.ac.uk/entry/Q8BNJ2. It was obtained by placing the peptide in the same initial position as found for docking to human ADAMTS4 allowing simultaneous contact of the Arg residues to the two homologous Glu residues (Glu397, Glu398) found in murine. The solvated structures were energy minimized (5000 steps) and after a heating phase (see above) served as start structures for additional unrestrained MD simulations in an explicit solvent at 310 K (37 °C) and a pressure of 1 bar.

**Complexation of $Gd^{3+}$ to the ADAMTS4-specific probe**. 260 mg of the DOTA-peptide (157 μmol, purchased with >95% purity from peptides&elephants GmbH (Hennigsdorf, Germany)) were dissolved in 15 ml ultrapure water. The pH was adjusted to 7.3 by adding 10% NaOH solution. Afterward, 85 mg $GdCl_3 \cdot 6\ H_2O$ (229 μmol) were dissolved in 500 μl ultrapure water and added in one portion to the peptide solution. The pH value was readjusted to 7.3, and the solution was stirred for 3 h at room temperature. The solution was diluted with ultrapure water to 30 ml total volume. The probe was desalted with an XK 70/28 column and Sephadex G-10 with an Äkta Pure 25 (GE Healthcare, now Cytiva Life Science). As eluent, a mixture of 5% EtOH and 0.1% AcOH in ultrapure water was used, and the

flowrate was 15 ml/min. The collected probes were lyophilized to obtain 150 mg of a white powder (83 μmol, 53% yield). The complete complexation of Gd to DOTA was examined via MALDI-ToF-MS: $[M + H]^+$ calculated 1806.64; found 1806.42 (Supplementary Fig. 16 and Supplementary Table 5).

**Complexation of $Eu^{3+}$ to the competition probe**. 250 mg of the DOTA-peptide (151 μmol, purchased with >95% purity from peptides&elephants GmbH (Hennigsdorf, Germany)) were dissolved in 15 ml ultrapure water. The pH was adjusted to 7.3 by adding 10% NaOH solution. Afterward, 88 mg $EuCl_3 \cdot 6\ H_2O$ (240 μmol) were dissolved in 800 μl ultrapure water and added in one portion to the peptide solution. The pH value was readjusted to 7.3, and the solution was stirred for 3 h at room temperature and overnight at 4 °C. The solution was diluted with ultrapure water to 30 ml total volume. The probe was desalted with an XK 70/28 column and Sephadex G-10 with an Äkta Pure 25 (GE Healthcare, now Cytiva Life Science). As eluent, a 60 μM ammonia acetate buffer was used, and the flowrate was 15 ml/min. The collected probes were lyophilized to obtain 164 mg of a white powder (91 μmol, 60% yield). The complete complexation of Eu to DOTA was examined via MALDI-ToF-MS: $[M + H]^+$ calculated 1801.64; found 1801.08 (Supplementary Fig. 17 and Supplementary Table 5).

**Complexation of $Gd^{3+}$ to the negative control probe**. 100 mg of the DOTA-peptide (57 μmol, purchased with >95% purity from peptides&elephants GmbH (Hennigsdorf, Germany)) were dissolved in 10 ml ultrapure water. The pH was adjusted to 7 by adding 10% NaOH solution. Afterward, 63 mg $GdCl_3 \cdot 6\ H_2O$ (169 μmol) were added in one portion to the peptide solution. The pH value was readjusted to 7, and the solution was stirred for 4 h at room temperature. The solution was diluted in the running buffer to 18 ml total volume. The probe was desalted with an XK 70/28 column and Sephadex G-10 with an Äkta Pure 25 (GE Healthcare, now Cytiva Life Science). As a running buffer, a mixture of 5% EtOH and 0.1% AcOH in ultrapure water was used, and the flowrate was 15 ml/min. The collected probes were lyophilized to obtain 49 mg of a white powder (26 μmol, 46% yield). The complete complexation of Gd to DOTA was examined via MALDI-ToF-MS: $[M + H]^+$ calculated 1894.63; found 1894.76 (Supplementary Fig. 18 and Supplementary Table 5).

**Metabolic stability**. 1 g of lyophilized human plasma was dissolved in 10 ml ultrapure water and mixed for 12 h at 4 °C. Afterward, the plasma was filtered with a pore size of 0.2 μm, and the filtrate was mixed for 1 h at 37 °C. 2 ml of plasma was mixed with 8 mg of the ADAMTS4-specific probe, while 2 ml plasma was kept blank. Both samples were shaken at 37 °C at 700 rpm. After different time points (5 min, 30 min, 45 min, 1 h, 2 h, 4 min, 18 h, and 24 h) respectively, 200 μl of both samples were removed and 200 μl guanidium HCl (6 M, pH 3) was added to each sample and mixed for 30 min at RT. Afterward, 1 ml EtOH was added, and the samples were shaken at 4 °C for 48 h. The samples were centrifuged at 4 °C, 22640 x g for 15 min, and 1 ml of the supernatant was taken. The solvent was removed with the Speedvac (Martin Christ RVC 2-18 CDplus) at 37 °C. 250 μl ultrapure water + 0.2% TFA were added into each sample and shaken overnight. The samples were centrifuged at 4 °C, 22640 x g for 15 min, and 200 μl were transferred into MS vials. The samples were measured with an AdvanceBio Peptide Map column (C18; Core-shell; length 15 cm; ID 2.1 mm) on a 1260 Infinity Bioinert Quaternary LC System (Agilent Technologies). The following conditions were used: 2 μl injections volume, flowrate 0.4 ml/min, 0–30 min 99% A to 85% A (A: ultrapure water with 0.2% TFA; B ACN with 0.16% TFA). Afterward, for each time point, the blank was subtracted from the sample spectrum.

**In vitro cytotoxicity testing**. Two different cell lines, C57BL/6 mouse primary aortic endothelial Cells (Cellbiologics, USA) and human aortic endothelial cells (ATCC®, USA), were used for in vitro cytotoxicity testing. C57BL/6 mouse primary aortic endothelial cells were purchased at Cellbiologics (C57-6052, Lot#M111718W10, Chicago, Illinois, USA) and human aortic endothelial cells were purchased by ATCC® (PCS-100-011, Lot#64408088, Manassas, Virginia, USA).

The cells grew up in T25 and T75 tissue culture flasks (Falcon™) with endothelial cell medium, which was supplemented with endothelial Cell Medium supplement Kit (0.5 mL VEGF, 0.5 mL ECGS, 0.5 mL heparin, 0.5 mL EGF, 0.5 mL hydrocortisone, 5.0 mL L-glutamine, 5.0 mL antibiotic-antimycotic solution and 25.0 mL FBS; Cellbiologics, USA). When cells were confluent, they were seeded in 6-well tissue culture plates or 96-well tissue culture plates. If cells were 80% confluent, they were used for experiments.

**MTT**. MTT assay is a method to quantify cell proliferation and vitality. The cells grew up in 96-well tissue culture plates and were added with different concentrations of the ADAMTS4-specific probe (0.3, 0.6, 0.8, 2.2, 4.4, and 8.8 μmol) for different times (3 h, 24 h, and 48 h). In addition, positive and negative control were examined. After 3 h, 24 h, and 48 h the media are discarded, and serum-free media and MTT (3-(4,5-dimethylthiazol-2-yl)-2,5-diphenyltetrazolium bromide) reagent (Abcam, UK) were added. The cells were incubated for 3 h at 37 °C. Then MTT solvent was added and shaken for 15 min. Finally, the absorption at 590 nm was measured on the photometer (PowerWave™ HT Microplate Spectrophotometer, USA).

The values were tested for significance by Anova (Excel, Version 16.57, Micrsoft®, Washington, USA).

**CytoTox-One™**. The CytoTox-ONE™ Homogeneous Membrane Integrity Assay is a fluorometric method for estimating the number of non-viable cells. In this assay, the release of lactate dehydrogenase (LDH) from cells with damaged membranes is measured. Cells were cultured in 96-well tissue culture plates and incubated with different concentrations of the ADAMTS4-specific probe (0.3, 0.6, 0.8, 2.2, 4.4, and 8.8 μmol) for different times (3 h, 24 h, and 48 h). After the time points, the assay plates were removed from the 37 °C incubator and cooled to 22 °C for 30 min. Then a volume of CytoTox-ONE™ reagent (Promega, Madison, United States) was added to the volume of cell medium in each well and shaken for 30 s, followed by incubation for 10 min at 22 °C. 50 μl of Stop Solution was added to each well and after 10 s of shaking, fluorescence was recorded at an absorbance of 560 nm. To calculate the cytotoxicity in percent, following formula was used:

$$\text{Precent Cytotoxicity} = 100 \times \frac{\text{Experimental} - \text{culture medium Background}}{\text{Maximum LDH Relase} - \text{Culture Medium Background}} \tag{1}$$

The results were tested for statistical significance using ANOVA (Excel, Version 16.57, Micrsoft®, Washington, USA).

**Celltiter-Glo**. The Celltiter-Glo 2.0 assay is a homogeneous method to determine the number of living cells in culture. It is based on the quantification of the ATP presence, an indicator of metabolically active cells. Cultivated cells were grown in 96-well tissue culture plates and incubated with different concentrations of ADAMTS4-specific probe (0.3, 0.6, 0.8, 2.2, 4.4, and 8.8 μmol) for different time points (3 h, 24 h, and 48 h). The plates were equilibrated for 30 min at room temperature. The same volume of Celltiter-Glo 2.0 as the volume of cell culture medium was added to each well and shacked for 2 min. To stabilize the luminescent signal, the plates were incubated at room temperature for 10 min, and then the luminescence was measured.

The significance was tested with ANOVA (Excel, Version 16.57, Micrsoft®, Washington, USA).

**Animal experiments**. For every in vivo experiment male apolipoprotein-E knockout (ApoE$^{-/-}$) (B6.129P2-ApoE$^{tm1Unc}$/J) mice at the age of 8 weeks were used.

All procedures and animal maintenance were performed according to the guidelines and regulations of the Federation of Laboratory Animal Science Associations (FELASA) and the local Guidelines and Provisions for Implementation of the Animal Welfare Act. The animal studies were approved by the local authorities (Lageso Berlin.) Mice were obtained from the Research Institute of Experimental Medicine at the Charité Berlin and maintained under barrier conditions, including a dark/light cycle of 12 h, an ambient temperature of 20 °C, and a humidity of 45%. A standard laboratory diet and water were provided *ad libitum*. For aneurysm induction, male 8-week-old apolipoprotein-E knockout (ApoE$^{-/-}$) (B6.129P2-ApoE$^{tm1Unc}$/J) mice underwent implantation of osmotic minipumps (Alzet model 2004, Durect Corp). Osmotic minipumps were implanted subcutaneously in the dorsal neck region and continuously delivered angiotensin II (AngII, Merck KGaA, Darmstadt, Germany) (1000 ng/kg/min) for up to 28 days ($n = 32$). Sham-operated mice ($n = 10$) served as the control group, receiving saline for 28 days instead of AngII via the osmotic minipumps.

For the cross-sectional study, MR imaging was performed after two and 4 weeks of AngII-infusion ($n = 10$ per group). At each time point, mice were scanned prior to and following the administration of the ADAMTS4-specific probe (0.13 mmol/kg body weight).

The longitudinal study includes an MR imaging session 1 week after aneurysm induction ($n = 12$) with the administration of the ADAMTS4-specific probe. Animals were then followed up for 4 weeks to evaluate the potential of the probes for the prediction of aneurysm progression. After 4 weeks, mice were sacrificed, and aortas excised.

For the early detection study, we used an AAA group ($n = 20$). MR imaging was performed after 2, 4, and 10 days of AngII-infusion. At each time point, mice were scanned prior to and following the administration of the ADAMTS4-specific probe (0.13 mmol/kg body weight). On day 28 post implantation, only native imaging was performed and subsequently, the aortic samples were excised.

**Anesthesia and euthanasia**. For minipump implantation, mice were anesthetized by an intraperitoneal injection of a combination of Medetomidine (500 μg/kg body weight), Fentanyl (50 μg/kg body weight), and Midazolam (5 mg/kg body weight). For recovery of the animals, anesthesia was antagonized using a combination of Atipamezole (2.5 mg/kg body weight), Naloxone (1200 μg/kg body weight), and Flumazenil (500 μg/kg body weight). For each imaging session, animals were anesthetized as described above. For an ex vivo analysis, mice were sacrificed and exsanguinated by arterial perfusion with saline. The aorta, including the right renal artery, was excised.

**In Vivo MR experiments**. MR imaging was performed using a clinical 3 T Siemens system (Biograph, Siemens Healthcare Solutions, Erlangen, Germany). The system was equipped with a 4-channel receive-coil array for mouse body applications (Mouse Heart Array, P-H04LE-030, Version 1, Rapid Biomedical, Germany). Following anesthesia injection, the animals were positioned in a prone position, and body temperature (37 °C) was monitored using an MR-compatible heating system (Model 1025, SA Instruments Inc, Stony Brook, NY) to avoid rapid cooling. For intravenous administration of the molecular probe, venous access was established using a small diameter tube with an attached 30 G cannula. The imaging protocol included the following sequences:

After a three-dimensional (3D) gradient-echo scout scan, a non-contrast-enhanced two-dimensional time-of-flight angiography (2D TOF) was acquired in transverse orientation for visualization of the abdominal aorta with the following parameters: field of view (FOV) of 200 × 200 mm, a matrix of 960 × 960, resolution of 0.2 × 0.2 × 0.5 mm, 40 slices, repetition time (TR)/echo time (TE) of 35 ms/4.44 ms, flip angle of 90°, and bandwidth of 124 Hz/Px. A maximum intensity projection (MIP) was then automatically generated to obtain an arterial angiogram of the abdominal aorta for planning the subsequent MR angiography and delayed-enhancement scans. A 2D inversion time (TI) scout sequence, planned perpendicular to the abdominal aorta, was used to determine the optimal TI for blood-signal nulling. Acquisition parameters were as follows: for the 2D TI sequence—FOV 340 × 340 mm, matrix 576 × 576, spatial resolution 0.6 × 0.6 × 3 mm, TR/TE 44.91 ms/2.09 ms, flip angle 35°, bandwidth 579 Hz/Px, TR between subsequent IR pulses 1000 ms; and for the high-resolution 3D IR FLASH sequence—FOV 57 × 57 mm, matrix 384 × 384, spatial resolution = 0.1 × 0.1 × 0.3 mm, 56 slices, TR/TE = 1019.72/7.02 ms, flip angle 30°, bandwidth 130 Hz/px, TR between subsequent IR pulses = 1000 ms.

**Competition experiment**. An in vivo competition experiment was performed in 3 ApoE$^{-/-}$ mice ($n = 3$) 4 weeks after AAA induction. On day 1, MR imaging was performed prior to and following the intravenous administration of the ADAMTS4-specific probe. On day 2, MRI was performed prior to and after the injection of a 7-fold higher dose of the non-paramagnetic europium-labeled ADAMTS4-specific probe. Then, a clinical dose of the ADAMST4 specific probe (0.13 mmol/kg) was administered with the following imaging after 20 min.

**Negative control probe**. In vivo MR imaging was performed in 6 ApoE$^{-/-}$ mice ($n = 6$): three mice ($n = 3$) received AngII through the osmotic minipumps, three mice ($n = 3$) received saline, serving as the control group. MR imaging was performed 4 weeks after minipump implantation. On day 1, a native MR scan was followed by the intravenous injection of the negative control peptide (0.13 mmol/kg) and a subsequent MR scan. 24 h later, mice were imaged prior to and following the intravenous administration of the ADAMTS4-specific probe (0.13 mmol/kg).

**Assessment of magnetic resonance imaging signal**. Analysis of resulting MR images was performed using OsiriX (version 7.1, OsiriX Foundation, Bernex, Switzerland). All morphometric measurements were conducted on high-resolution MRI images. In order to assess the signal intensities, regions of interest (ROIs) were measured as areas of signal enhancement, which were co-localized with areas of aneurysmal-aortic tissue. For these areas, the CNR was calculated using the following formula: CNR = (Combined vessel-wall and aneurysmal-aortic-tissue-signal − blood-signal)/standard-deviation-signal (noise). The signal of noise was defined as the standard deviation in pixel intensity from an ROI placed in the background air anterior to the aorta.

**Relaxivity of the ADAMTS4-specific probe at 3 Tesla**. One milliliter of a 1 μg/ml solution of the ADAMTS4-specific probe in 1% BSA in PBS buffer was measured in a reaction tube using a 3 T MRI scanner. 1% BSA in PBS buffer was used as a negative control. After each scan, 50 μl of a 0.2 μg/μl solution of ADAMTS4 (R&D systems, 4307-AD-020) was added to one reaction tube, while 50 μl of the 1% BSA in PBS buffer was added to the second reaction tube. Tubes were shaken and measured again. The following ADAMTS4 concentrations were measured: 0 μg, 10 μg, 20 μg, 30 μg, 40 μg, 50 μg, and 60 μg.

**Histological analysis of the aortic aneurysms and aortic aneurysm morphometry**. Histological analyses were performed for correlation of in vivo and ex vivo data. Following surgical excision, aortic aneurysms tissues were frozen at −20 °C until further preparation. Tissues were then embedded in OCT compound and cut at −20 °C into 10 μm cryosections and immediately mounted on SuperFrost Plus adhesion slides (Thermo Scientific). These sections were stained for Miller's Elastica van Gieson stain (EvG) and Hematoxylin and Eosin (HE) staining. For morphological measurements, histological slices were examined and photographed using a light microscope (BZ-X800, Keyence, Osaka, Japan). Digitized images of EvG sections were used for morphological measurements using computer-assisted image analysis (ImageJ software, Version 1.51).

**Immunohistological analysis**. Immunofluorescence staining was performed to visualize ADAMTS4 within the aortic wall and evaluate the co-localization with

macrophages. Sections were incubated with polyclonal ADAMTS4 antibody (Rabbit anti-mouse ADAMTS4, PA1-1749A, Invitrogen 1:100 dilution) and monoclonal CD68 antibody (Rat anti-mouse CD68, clone FA-11, MCA1957, Bio-Rad, 1:100 dilution) overnight at 4 °C, each diluted in Dako REAL Antibody Diluent (Dako, Denmark). Following a two-time wash with PBST (0.05% Tween-20, pH 7.4), slides were incubated with polyclonal secondary antibody Alexa Fluor 647 (Donkey anti-rabbit IgG, A-31573 Thermo Fisher Scientific, Germany, 1:200 dilution) and Alexa Fluor 568 (Goat anti-rat IgG, A-11077, Thermo Fisher Scientific, Germany, 1:200 dilution) for 1 h at room temperature. Counterstaining and mounting were performed using Roti®-Mount FluorCare (Carl Roth, Germany). For the quantification of ADAMTS4 and CD68, digitized images of immunofluorescence sections were analyzed using a computer-assisted image analysis (Keyence BZX-800 Analyzer, Osaka, Japan). For visualization of smooth muscle cells (SMC), slides were incubated with anti-smooth muscle actin, rabbit-anti-mouse primary antibody, 1:400 dilution (E-AB-34268.20, Elabscience, TX, USA) for 90 min and were then washed three times with PBS-Tween. Sections were then incubated with a secondary goat anti-rabbit antibody Alexa Fluor 488, 1:500 dilution (ab10085, Abcam, Cambridge, UK) for 60 min. For visualization of ADAMTS4, slides were incubated with an anti-ADAMTS4 rabbit-anti-mouse antibody, 1:100 dilution (ab185722, Abcam, Cambridge, UK) overnight at 4 °C. Sections were then washed three times with PBS-Tween and incubated with a secondary Donkey anti-mouse antibody Alexa Fluor 568 for 60 min (A10042, Thermo Fisher Scientific, MA, USA, 1:200 dilution). For visualization of CD68, slides were incubated with an anti-CD68 rat anti-mouse antibody, 1:100 dilution (MCA1957GA, Bio-Rad, Feldkirchen, Germany) overnight at 4 °C. Sections were then washed three times with PBS-Tween and incubated with a secondary goat anti-rat antibody Alexa Fluor 568 for 60 min (A11077, Thermo Fisher Scientific, MA, USA, 1:200 dilution). All slides were then counterstained and mounted using Roti®-Mount Flour Care (Carl Roth, Germany). The sections were then scanned and analyzed using a digital microscope (Keyence BZX-800 Viewer and Analyzer, Osaka, Japan).

For visualization of ADAMTS4 and co-localization with macrophages within the human aneurysmal wall, immunohistological analysis of ruptured human aortic aneurysm was performed. Slides were incubated with polyclonal ADAMTS4 antibody (Rabbit anti-mouse ADAMTS4, PA1-1749A, Invitrogen 1:100 dilution) and monoclonal CD68 antibody (Rat anti-mouse CD68, clone FA-11, MCA1957, Bio-Rad, 1:100 dilution) overnight at 4 °C, and diluted in Dako REAL Antibody Diluent (Dako, Denmark). Following a two-time wash with PBST (0.05% Tween-20, pH 7.4), slides were incubated with polyclonal secondary antibody Goat anti-rabbit IgG:HRP (32460, Invitrogen 1:100 dilution) and polyclonal secondary antibody Rabbit anti-rat IgG:HRP (STAR21B, Bio-Rad, 1:200 dilution) for 1 h at room temperature. Sections were analyzed using ImageScope_64_v12.4.0.5043 (Leica, Germany)

**Fluorescein-labeled probe (FLP).** In order to prove the specific binding of the ADAMTS4-specific probe, the probe was purchased with a 5-carboxyfluorescein label instead of the DOTA complex at the lysine side-chain amine (peptides&elephants GmbH, Hennigsdorf, Germany) and administered intravenously in one ApoE$^{-/-}$ mouse 4 weeks after AAA induction and one sham-operated mouse (0.13 mmol/kg). Mice were euthanized 20 min after administration, and aortic tissue was harvested for histological analysis as described above.

**Inductively coupled plasma mass spectroscopy (ICP-MS).** Digestion of tissue, blood, and urine samples for biodistribution:

In order to analyze the biodistribution of the ADAMTS4-specific probe, blood and urine samples, as well as tissue samples (heart, thoracic aorta, lungs, liver, spleen, kidney, muscle) were collected 15, 30, 60, and 120 min after the probe injection ($n = 3$ per time point). Samples were prepared, and gadolinium concentrations were measured according to the ICP-MS protocol.

Digestion of samples: To measure the local gadolinium concentration in the abdominal aortic wall, ICP-MS was performed. For sample preparation, aortic tissues ($n = 3$ per group for sham, 2 weeks, and 4 weeks old mice) were dried at 39 °C overnight. Afterward, 1 ml of nitric acid (Carl Roth, 68%, puriss.) was added, and the samples were kept overnight at room temperature. On the next day, 200 μl of hydrogen peroxide (Carl Roth, 30%, p.a.) was added, and the mixture was shaken at room temperature until no further gas formation was detected. The samples were dissolved with 12 ml ultrapure water. For the liver samples, a double amount of nitric acid, hydrogen peroxide, and ultrapure water for dilution was used.

Analysis of digested samples by ICP-MS: Digested samples were diluted in HNO$_3$ (purified by sub-boiling distillation s.b.), 1% and measured with the iCAP Qc ICP quadrupole mass spectrometer (Thermo Fisher Scientific, Bremen, Germany) in combination with the autosampler 4DXF-73A (ESI Elemental Service & Instruments GmbH, Mainz, Germany) using a 200 μL PFA nebulizer and a cyclonic spray chamber. Calibrations were carried out using diluted gadolinium ICP Standard CertiPUR (Merck KGaA, Darmstadt, Germany).

**Element specific bioimaging using laser ablation (LA) coupled to inductively coupled plasma mass spectrometry (LA-ICP-MS).** Aortic tissues were cut into 10 μm cryosections at −20 °C and mounted on SuperFrost Plus adhesion slides (Thermo Scientific, Waltham, MA, USA). Laser ablation inductively coupled plasma mass spectrometry (LA-ICP-MS) analysis was performed on a commercial LA system (NWR-213, ESI, Bozeman, MT, USA) equipped with a two-volume sample chamber coupled to a sector field ICP-MS (Element XR, Thermo Fisher Scientific, Bremen, Germany) (Supplementary Table 6).

Helium was used as the carrier gas, and argon was added before reaching the torch using a Y-piece (Supplementary Table 7). The ICP-MS was tuned for maximum ion intensity and good signal stability (RSD < 5%), keeping the oxide ratio (ThO/Th) below 1% during the continuous ablation of a microscopic glass slide. Matrix-matched agarose gel standards cast on glass slides were used for drift control and calibration[67]. These standards contain Gd concentrations between 26.6 pg mm$^{-2}$ and 600.1 pg mm$^{-2}$. Averaged intensities of six line-scans per standard were used for calibration. Data visualization was done in Origin 2022 (OriginLab Corporation, Northampton, MA).

**Statistical analysis.** Values are specified as mean ± standard error of the mean. For comparison of continuous variables, a Student's t-test (unpaired, two-tailed) was applied. Linear regression was used to determine the relationship between in vivo and ex vivo measurements. A $P$-value < 0.05 was considered to indicate a statistically significant difference.

**Reporting summary.** Further information on research design is available in the Nature Research Reporting Summary linked to this article.

## Data availability

All data generated in this study are provided in the Supplementary Information/Source Data file. The raw data that support the findings of this study has been deposited in the following link: https://figshare.com/projects/NCOMMS-21-00839/138081

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

## Acknowledgements

This study was funded by the Deutsche Forschungsgemeinschaft (DFG, German Research Foundation)—Project-ID 372486779—SFB 1340 2018; MA 5943/3-1/4-1/9-1 and the BHF program grant—RG/12/1/29262. J.O.K. thanks Dr. Teresia Hallström from NanoTemper and Dr. Zoltán Konthur from BAM 1.8 for assistance with MST measurements. L.C.A. is grateful for her participation in the BIH Charité-Clinician Scientist Program funded by the Charité-Universitaetsmedizin Berlin and the Berlin Institute of Health.

## Author contributions

J.O.K. performed and analyzed all experiments on the synthesis and ex vivo evaluation of the ADAMTS4-specific probe. J.B. performed and analyzed the animal and ex vivo experiments. J.O.K. and J.B. wrote the manuscript. M.R.M. and M.G.W. designed and supervised the study. A.K. performed and analyzed the in vitro experiments. J.S. and H.T. performed the LA-ICP-MS and ICP-MS experiments. D.B.M. performed the negative control and early detection animal experiments. M.Z. performed and analyzed the

docking experiments. W.E.K. and L.M. performed the humane tissue staining. M.P. and M.G.W. participated in ex vivo experiments and reviewed the manuscript. T.S., L.C.A., M.P., J.M., R.M.B., M.T., and B.H. reviewed and commented on the manuscript.

## Funding

## Competing interests

The authors declare no competing interests.
