## [Peer Review File · Nature Communications]

Reviewers' Comments:

Reviewer #1:

Remarks to the Author:

The authors have identified a cyclic peptide candidate carrying a highly specific targeting potential towards ADAMTS4 (A disintegrin and metalloproteinase with thrombospondin motifs 4) of which have been suggested to mediate abdominal aortic aneurysms (AAA). With the reported results, it has been further shown that peptide is also capable of acting as a MR probe and could be used to detect AAA in mouse models. Further applications for early detection in clinical settings involving ADAMTS4 were suggested to be possible.

For peptide identification, they used a one-bead-one-compound libraries (OBOC) combinatorial method of which was used to select positive hits for binding ADAMTS4. The peptide was computationally tested for binding, analyzed for affinity using microscale thermophoresis. Furthermore, its suitability as a MR probe, its metabolic stability, its cytotoxicity at cell culture level and biodistribution were clearly reported. These steps were followed by the observations on the implementation of the ADAMTS4-specific probe for in vivo MRI. They have also reported ex vivo expression ADAMTS4 in the aortic wall in order to display that for binding involves ADAMTS4. All of these results were considered as a significant contribution to the literature due to the identification of a novel peptide based MR probe for testing ADAMTS4 involvement in AAA. Results reported are complementary of each other and designed in a sequential order presenting the story behind the study. The OBOC methodology and design strategy taken for selection of small peptide in the context of ADAMTS4 is novel and other methods used such as its application in MRI validates the hypothesis very clearly. The methods were explained in detail and more than sufficient for reproduction, if needed.

Some minor concerns encountered when reading, could be found as below. After handling these minor issues, I recommend acceptance of the manuscript.

- 1) Using entropy term, rather than entropy may be a more preferable when denoting its effect on Gibbs free energy.
- 2) When figure 1 is investigated alone, the role antibodies should be explained more clearly in the legend. It is much better for readers when every figure legend is comprehensible with minimal referencing to the text.
- 3) In the methods section, "Screening against ADAMTS4", why do the authors use the anti-his antibody before the actual screening? Is to check for non-specific binding of the antibody to the beads with peptides? This should be explained more clearly in the methods as done in supplementary figure 1.
- 4) The peptide identified is a very short one, carrying probable non-specific binding potential. While the authors discussed the possible non-specific binding of the peptide to other targets, not analyzed in the study, it would be much helpful for readers to see what sort of available methods do exist for analyzing the non-specific binding.
- 5) One major concern is the use of human ADAMTS4 in docking and MD studies rather than its mouse counterpart. While the salt bridges identified with the human ADAMTS4 does explain the binding, the location of the Glu residues on the mouse counterpart should be shown. If similar counterparts for the Glu residues do not exist on mouse ADAMTS4, at least a global docking study should be implemented to show probable target regions. While this concern may not affect major conclusions of the study, I believe the results in mouse ADAMTS4 will strengthen the overall data for the computational perspective.
- 6) Conformational stability analyses of replicates for MD simulations should be added to supplementary figures section.
- 7) MTT assay is informative for cytotoxicity but more advanced methodologies do exist. For such an all-encompassing study, more detailed evaluation on cytotoxicity may be completed for necrosis or apoptosis markers using western blotting. Moreover, confirmation of the MTT findings may be validated with other methods ie. LDH release assay or CytoTox-Glo. If authors do not prefer to add these due to funding, a related section should also be added to the discussion to make readers aware of potential toxicity more profoundly.

Reviewer #2:

Remarks to the Author:

The paper by Kaufmann et al. describes the synthesis of a one-bead-one-compound (OBOC) combinatorial library of cyclic peptides and screen it against ADAMTS4 resulting in the identification of a strong and highly selective binder. Based on the initial screening results, a probe was developed, characterized and its properties as an in vivo probe for prediction of aneurysm expansion demonstrated in an abdominal aortic aneurysm mouse model. The paper is very comprehensive and very nicely demonstrate the full value of the OBOC methodology.

However, the paper may be more appropriate for a more general journal. The chemistry part is very sparse and does not contain novel chemistry elements that can justify publication in Nat Chem.

The paper would benefit for a general proofreading as several passages are difficult to read, see examples in the attached file.

I am missing relevant references for important parts, including previous work aiming at developing molecular probes for ADAMTS4 (see for example ACS Appl Mater Interfaces. 2013 Jul 10;5(13):6089-96).

I find the title misleading as the method allowing identification of the ADAMTS4 probe is not 'fully-synthetic', it involves screening in addition to med chem optimization.

It may be relevant to include a study with mutant enzymes to validate their hypothesis regarding the binding mechanism.

Reviewer #3:

Remarks to the Author:

The authors applied innovative OBOC approach to synthesize the library of cyclic peptide with the intention to identify the specific binder for ADAMTS4. Assumed specific probe was then used in a series of in vivo experiments to demonstrate its utilization in the AAA field. The lack of specific probe namely does not allow timely diagnosis of potentially fatal condition.

Although the work is of significance in the OBOC field, the study itself suffers some methodological weaknesses. The full list of comments can be find in the annotated manuscript. Here only the main points are listed.

1. The description of the library synthesis would benefit from more detailed description of the library QC process.
2. The major weakness represents the biophysical characterization of the peptides either with SPR or MST. The way how the SPR data are reported does not depict current SPR standards. Please, see annotated manuscript for more details.
3. The use of RED-tis-NTA dye for the MST requires a standard pre-test, which determines the affinity of the dye to the his-tag of the protein. This value should be reported, because the entire experimental design afterwards depends on this number. Also, for the optimal MST window I recommend to label the peptides with the fluorophore like Cy5 or disulfo-Cy5 and use these in the MST assay. I am aware of the fact that ADAMTS variant are not simple to handle and are very precious. However, the data would be most likely much more reliable. Currently, the MST amplitude below 2 provides a hint about the Kd, but no reliable value.
4. Careful observation of the MST data suggests that the same sample was measured three times. I recommend three biological replicates instead.
5. Significant revision of the SPR and MST data/experiments is required to unambiguously demonstrate the nanomolar binding affinity and specificity of ADAMTS4 specific probe. Current data do not support this assumption with a doubt.
6. Fig 4b: only the minor portion of the staining with antibody and the probe overlap. Thus, this image is more indicative of lack of specificity than the proof of specificity. Please, adjust the interpretation accordingly.
7. in vivo specificity of the probe - using the same probe in "hot" and "cold" version does not underscore the specificity of the probe. It is expected that they will bind to the same binding

crevice.

8. In vivo MR imaging - statistical analysis of the data is needed to support the statements.

General observation:

when reading the manuscript the writing styles between different parts of the manuscript vary significantly. Some are easy to understand, in some some improvements are recommended.

Reviewer #4:

Remarks to the Author:

This paper by Kaufmann et al. describes the development of an ADAMTS4-specific MR probe, which they then tested to visualize AAA in an ApoE^{-/-}+Ang II model.

Strength: A marker for early detection of AAA or TAA is one of the deficiencies in the medical field, therefore, this study is a step in the right direction.

Weaknesses:

1) When are ADAMTS4 levels elevated in an aneurysmal aorta? How early before the development of AAA?

2) Related to the previous point, the authors show that ADAMTS4-MR probe labeled the aneurysmal aorta (at 2wk and 4wks), however in both time points there is a visible AAA which is also detected by MRI or echo-ultrasound (echo would give more clear images than the shown MRI images), therefore, using this probe would not really help in early detection of AAA.

3) It is also important to know which cell types in the aorta express ADAMTS4 when undergoing AAA development, this would further help in identifying the type of aneurysm. E.g. the ApoE^{-/-}+AngII model (used in this study) causes an outward remodeling where the outer diameter of AAA is greater than the lumen dilation (and is associated with plaque deposition and metabolic alterations). Would ADAMTS4 probe labeling also work where there is aortic wall degradation and lumen dilation without the outward remodeling?

Reviewer #5:

Remarks to the Author:

This manuscript reports the identification of a novel cyclic peptide with a high affinity to ADAMTS4 through screening of a one-bead-one-compound library. The peptide was then modified for development of an MR probe through conjugation of DOTA-gadolinium. Proof-of-concept validations determined the potential of this probe for detection of abdominal aortic aneurysm using a mouse model. Molecular imaging of key processes involved in the pathogenesis of aortic aneurysm is a highly needed strategy to improve the management of patients. The approach and targeted molecule are novel, and the experimental approach is robust. I only have a few comments, as summarized below.

1. Figure 2b: There is a marked variation in % binding both within each concentration and between different concentrations. It is also unclear if the binding has reached the plateau. These affect K_d measurement. Increasing N may help to resolve the issue. The same issues apply to Supplemental Figure 2.

2. Figure 4c: Representative images from mice with blocking experiments are required.

3. Please explain the rationale for using a sevenfold dose of non-labeled probe in blocking experiments. This dose seems very low to achieve saturation of the target molecule (ADAMTS4) in tissues. Therefore, it is surprising that ~80-90% blocking is achieved with this dose. Biodistribution data from blocking experiments will also be helpful.

4. Figure 4d-e: The increase in CNR after injection of the negative control probe seems visually a

lot more than the quantified values in panel D. Perhaps panel E may be replaced by a more representative image. Also, it is ambiguous to which groups the reported P values refer to (panel D).

Point-by-point response to these comments NCOMMS-21-00839A-Z

All changes in the manuscript text file are also indicated by colour highlighting.

Reviewer #1:

The authors have identified a cyclic peptide candidate carrying a highly specific targeting potential towards ADAMTS4, a disintegrin and metalloproteinase with thrombospondin motifs 4) of which have been suggested to mediate abdominal aortic aneurysms (AAA). With the reported results, it has been further shown that peptide is also capable of acting as a MR probe and could be used to detect AAA in mouse models. Further applications for early detection in clinical settings involving ADAMTS4 were suggested to be possible.

For peptide identification, they used a one-bead-one-compound libraries (OBOC) combinatorial method of which was used to select positive hits for binding ADAMTS4. The peptide was computationally tested for binding, analyzed for affinity using microscale thermophoresis. Furthermore, its suitability as a MR probe, its metabolic stability, its cytotoxicity at cell culture level and biodistribution were clearly reported. These steps were followed by the observations on the implementation of the ADAMTS4-specific probe for in vivo MRI. They have also reported ex vivo expression ADAMTS4 in the aortic wall in order to display that for binding involves ADAMTS4. All of these results were considered as a significant contribution to the literature due to the identification of a novel peptide based MR probe for testing ADAMTS4 involvement in AAA. Results reported are complementary of each other and designed in a sequential order presenting the story behind the study. The OBOC methodology and design strategy taken for selection of small peptide in the context of ADAMTS4 is novel and other methods used such as its application in MRI validates the hypothesis very clearly. The methods were explained in detail and more than sufficient for reproduction, if needed.

We thank the reviewer for this thorough analysis of our study.

Some minor concerns encountered when reading, could be found as below. After handling these minor issues, I recommend acceptance of the manuscript.

1) Using entropy term, rather than entropy may be a more preferable when denoting its effect on Gibbs free energy.

We thank the reviewer for this comment and agree. Changes were made accordingly throughout the manuscript.

“We adapted the applied on-chip screening technique to the requirements of in vivo MR imaging with an emphasis on cyclic peptides. Due to the more rigid structure, cyclic peptides have large surface areas and decreased entropy terms of the Gibbs free energy, which favoured high and selective binding. An additional favourable characteristic of the cyclic structure is the higher hydrolysis stability against protease activities compared to linear peptides.^{21,22”}

2) When figure 1 is investigated alone, the role antibodies should be explained more clearly in the legend. It is much better for readers when every figure legend is comprehensible with minimal referencing to the text.

We agree with the reviewer and adapted the figure legend accordingly.

“For the screening, we used our target protein, which was detected by an anti-His-Ab. The anti-His-Ab was afterwards stained with an anti-Mouse IgG-Atto 633 antibody. After immobilization of the beads on a chip, firstly, a pre-scan against both antibodies was performed, to exclude positive hits against the antibodies. This was followed by the main scan against ADAMTS4 and a second scan against ADAMTS5.”

3) In the methods section, “Screening against ADAMTS4”, why do the authors use the anti-his antibody before the actual screening? Is to check for non-specific binding of the antibody to the beads with peptides? This should be explained more clearly in the methods as done in supplementary figure 1.

The screening design we used is setup with a pre-scan against the anti-his antibody and the according to staining antibody. The pre-scan was performed, since both antibodies could also work as a target, and thus, all beads, which demonstrate binding against the antibodies, were identified. Based on this approach, the number of false-positive hits in the subsequent screenings was reduced and positive hits from the pre-scan were excluded in the following screenings. Additionally, the non-specific binding of the antibodies to the beads was characterised. This non-specific binding led to a small increase of the fluorescence, which was eliminated from the main screening.

We added a paragraph to the manuscript specifically describing this approach to the reader.

“This prescan was performed, since both antibodies could also work as a target, and thus, all beads, which demonstrate binding against the antibodies, were identified. Based on this approach, the number of false positive hits in the subsequent screenings were reduced and positive hits from the prescan were excluded in the following screenings. Additionally, the non-specific binding of the antibodies to the beads was characterized. This non-specific binding led to a small increase of the fluorescence, which was eliminated from the main screening.”

4) The peptide identified is a very short one, carrying probable non-specific binding potential. While the authors discussed the possible non-specific binding of the peptide to other targets, not analyzed in the study, it would be much helpful for readers to see what sort of available methods do exist for analyzing the non-specific binding.

This is an important comment of the reviewer. Non-specific binding can be a relevant challenge for OBOC screenings as well as for nearly all other screenings, including phage display or ribosomal screening. To address this issue, several adaptations of the used technique were performed: First, the used buffer was mixed with detergent (Tween-20) to reduce unspecific binding. Additionally, the use of blocking proteins (e.g. cas-BSA) in the buffer was used to block non-specific binding sites.

The screening against different targets can reveal non-specific binding, since positive hits, which occur for all targets, might be a sign for a non-specific binder. In our case, we used two different additionally scans. First, we screened against both antibodies, to neutralize all positive hits, which

bind the antibodies. Subsequently, we performed the screening against ADAMTS5, which is structural comparable to ADAMTS4. Binders, which show a similar binding against ADAMTS4/5 were treated as not selective. We added the following paragraph:

“For such a low molecular weight peptide the high specificity against ADAMTS4 was surprising. To clarify the specificity of our probe, we performed MST measurements against structurally comparable proteins such as ADAMTS1 and ADAMTS5, which demonstrated lower binding constants. We also performed in vivo competition experiments with a “cold” probe to confirm that there was no relevant unspecific binding.”

5) One major concern is the use of human ADAMTS4 in docking and MD studies rather than its mouse counterpart. While the salt bridges identified with the human ADAMTS4 does explain the binding, the location of the Glu residues on the mouse counterpart should be shown. If similar counterparts for the Glu residues do not exist on mouse ADAMTS4, at least a global docking study should be implemented to show probable target regions. While this concern may not affect major conclusions of the study, I believe the results in mouse ADAMTS4 will strengthen the overall data for the computational perspective.

We thank the reviewer for this important comment. We specifically used human-ADAMTS4 in the docking and MD studies since the screening and biophysical measurements were performed with human-ADAMTS4. This is of high importance as the overarching aim of this study is to provide to groundwork for a translation of the probe into a human setting. One aim of this study was to confirm, that the binding pocket, which we detected in human ADAMTS4, also exists in mouse-ADAMTS4. We could demonstrate that both ADAMTS4 species have the same binding cleft and both contain the two glutamine acid residues at the binding site. While in human-ADAMTS4, they are located on positions 401 and 402, in mouse-ADAMTS4 they are located on positions 397 and 398. This shift is a result of the shorter peptide chain in the flexible start region of mouse-ADAMTS4. The differences in the peptide chain occur in the starting region and were not part of the MD docking-study or they occurred at a distant location relative to the actual binding cleft and did not influence the binding site. For this revision, we performed additional experiments with the simulation using mouse-ADAMTS4 and we could confirm that the binding remains stable (Response to the reviewer figure 1).

Response to the reviewer figure 1: Simulation snapshot from the simulation (~25 ns) of the mouse AdamTS in complex with peptide binder. Mouse AdamTS in orange cartoon, Glu397,398 as sticks, peptide as stick model including hydrogens.

6) *Conformational stability analyses of replicates for MD simulations should be added to supplementary figures section.*

As suggested by the reviewer, the conformational stability analysis of replicates for MD simulations was performed and results were added to the supplementary figures section (Response to the reviewer figure 2).

Additionally, we performed a computational mutation study by exchanging the Glu amino acids against Gln. No binding could be measured for the mutated ADAMTS-4. This is consistent to our MST alanine walk experiments, where we did not measure any binding by the respective exchange of the Arg amino acids by alanine.

Response to the reviewer figure 2: Root-mean-square deviation (RMSD of non-hydrogen atoms) vs. simulation time for three separate simulations of AdamTS4 in complex with peptide binder (A-C). Simulations were started with different initial velocity distributions. The RMSD of the complete protein vs. simulation time is indicated in black; the RMSD of the peptide binder with respect to the start placement (after best superposition of the protein on the start structure) is shown in red. (D) same simulation for the mouse ADAMTS4 in complex with the peptide binder (same initial position as in human AdamTS4 cases). (E) same as in (A) but with the in silico human AdamTS4 Glu401Gln, Glu402Gln mutation. Note the RMSD scale is different in (E).

7) *MTT assay is informative for cytotoxicity but more advanced methodologies do exist. For such an all-encompassing study, more detailed evaluation on cytotoxicity may be completed for necrosis or apoptosis markers using western blotting. Moreover, confirmation of the MTT findings may be validated with other methods ie. LDH release assay or CytoTox-Glo. If authors do not prefer to add these due to funding, a related section should also be added to the discussion to make readers aware of potential toxicity more profoundly.*

We thank this reviewer for this comment. To confirm the MTT findings, we performed an additional Celltiter-Glo assay and a CytoTox-One assay. The CellTiter-Glo assay is a method of determining the number of viable cells based on the quantification of ATP present. The CytoTox-One assay measures the release of LDH from damaged cell membranes.

With these additional measurements, we could confirm the results of the MTT assay. The human and mouse aortic endothelial cells have no significant change by our ADAMTS-4 probe. Our results

were statistically analysed, and there was no significant difference ($p > 0.05$) between the negative control and all concentration at all time points. The additional experiments, an additional supplementary figure and derived conclusions were added to the manuscript (Response to the reviewer figure 3).

Response to the reviewer Figure 3: Cytotoxicity test on human aortic endothelial cells (HAEC) and mouse aortic endothelial cells (MAEC). a, MTT- assay with (left) HAEC and MAEC (right) on different time points (3 h, 24 h, 48h) and different concentrations of ADAMTS-4 probe (0.3, 0.6, 0.8, 2.2, 4.4 and 8.8 μmol). The difference between the concentrations and the negative control was not significant ($p > 0.05$). A significant difference was found between the positive control and the concentrations ($p < 0.05$). pc = positive control; nc = negative control. b, Cytotox-One assay with (left) HAEC and MAEC (right) on different time points (3 h, 24 h, 48 h) and concentrations of ADAMTS-4 probe (0.3, 0.6, 0.8, 2.2, 4.4 and 8.8 μmol). The percentage of cytotoxicity was determined. No significant difference was found between the concentrations and the negative control ($p > 0.05$). c, Celltiter-Glo assay with (left) HAEC and MAEC (right) on different time points (3 h, 24 h, 48 h) and concentrations of ADAMTS-4 probe (0.3, 0.6, 0.8, 2.2, 4.4 and 8.8 μmol). There was no significant difference between the concentrations and the negative control ($p > 0.05$). Significant difference was found in positive control and concentrations ($p < 0.05$). PC = positive control, NC= negative control, RLU= relative light unit.

Reviewer #2 (Remarks to the Author):

The paper by Kaufmann et al. describes the synthesis of a one-bead-one-compound (OBOC) combinatorial library of cyclic peptides and screen it against ADAMTS4 resulting in the identification of a strong and highly selective binder. Based on the initial screening results, a probe was developed, characterized and its properties as an in vivo probe for prediction of aneurysm expansion demonstrated in an abdominal aortic aneurysm mouse model. The paper is very comprehensive and very nicely demonstrate the full value of the OBOC methodology.

We thank the reviewer for the appreciation of our study.

However, the paper may be more appropriate for a more general journal.

The chemistry part is very sparse and does not contain novel chemistry elements that can justify publication in Nat Chem.

We agree with the reviewer. This study was however submitted to “Nature Communications” and this is currently the revision process of “Nature Communications” we are responding to. The study was not submitted to “Nature Chemistry”.

We apologize if this was not made clearer by us.

The paper would benefit for a general proofreading as several passages are difficult to read, see examples in the attached file.

We apologize and improved the readability throughout the manuscript.

I am missing relevant references for important parts, including previous work aiming at developing molecular probes for ADAMTS4 (see for example ACS Appl Mater Interfaces. 2013 Jul 10;5(13):6089-96).

We agree with the reviewer and apologize for this oversight; the according publication was added to the manuscript:

“Peng et al. developed a method for ex vivo testing of ADAMTS4, which shows high sensitivity.⁵⁶ While this approach shows great potential for ex vivo ADAMTS4 detection, the use of an in vivo MR imaging agent provides in vivo spatial information regarding the distribution within the vessel wall.”

I find the title misleading as the method allowing identification of the ADAMTS4 probe is not ‘fully-synthetic’, it involves screening in addition to med chem optimization.

We agree with the reviewer and updated the title of our study, which now reads:

“Identification and development of an ADAMTS4-specific MR-probe to assess aortic aneurysms in vivo using synthetic peptide libraries”.

It may be relevant to include a study with mutant enzymes to validate their hypothesis regarding the binding mechanism.

We thank the reviewer for this comment. Mutation studies are of high importance to validate the binding-mechanism. We performed an MD-simulation to address this issue (Response to the reviewer figure 4). The simulation demonstrated a fast peptide break from the position by mutation one of the glutamine acids in the binding cleft. This is in line with our hypothesis regarding the binding cleft and the exchange of arginine by alanine in the MST experiments.

We adapted the manuscript accordingly and added the additional figure.

Response to the reviewer figure 4: Simulation snapshots of the in silico double mutation (Glu401Gln, Glu402Gln) at early simulation stage (A, ~2 ns) and upon complete dissociation of the peptide (B, ~25 ns). Human AdamTS4 in orange cartoon, mutated residues as sticks, peptide as stick model including hydrogens.

Reviewer #3 (Remarks to the Author):

The authors applied innovative OBOC approach to synthesize the library of cyclic peptide with the intention to identify the specific binder for ADAMTS4. Assumed specific probe was then used in a series of in vivo experiments to demonstrate its utilization in the AAA field. The lack of specific probe namely does not allow timely diagnosis of potentially fatal condition. Although the work is of significance in the OBOC field, the study itself suffers some methodological weaknesses. The full list of comments can be find in the annotated manuscript. Here only the main points are listed.

1. The description of the library synthesis would benefit from more detailed description of the library QC process.

As suggested by the reviewer we added a detailed description of the library QC process to the manuscript:

“For quality control of the library, we used a peptide with defined sequence, which we synthesized simultaneous to the library as control. With this peptide, we could examine the duration of the cyclisation and the truncation sequence in the MALDI-TOF-MS. Additionally, we developed test-chips with the library and examined them with MALDI-TOF-MS, to test, if we can measure the truncation also under screening conditions. In contrast to the test-peptide, we could not detect the truncation sequences on the chip. Therefore, we tested the alkylation of the free cysteine in the truncation sequence. We used N-ethylmaleimide (NEM) and iodoacetamide (IA) for alkylation. We saw highly double alkylated peptides with NEM and focused more on IA in the MALDI-TOF-MS. By checking four different timepoints (15 min, 30 min, 45 min and 60 min) we saw best results after 30 min of IA.”

2. The major weakness represents the biophysical characterization of the peptides either with SPR or MST. The way how the SPR data are reported does not depict current SPR standards. Please, see annotated manuscript for more details.

We agree that demonstrating all SPR data is relevant for the reader. We attached all the missing data in the supplementary material (Response to the reviewer figure 5). The SPR data were focused on the classification, which of the five peptides is the most promising candidate.

Response to reviewer figure 5: *a*, Represents the immobilization of the anti-His-tag ab on a hydrogel gold chip. The chip was NHS activated twice. At the second time, some air was injected unintentional, too. The antibody was injected over both channels, even though most of it was bound on the sample channel. Afterwards the chip was quenched by aminoethanol. *b*, Shows the capturing of ADAMTS4. ADAMTS4 was injected over the sample channel for 15 min. Afterwards, the diverter was opened and buffer was flushing over both channels, until the baseline was nearly stable. The amount of captured ADAMTS4 was calculated for 26% of the R_{max} . Represents the SPR spectra of the 5 peptides from the screening. The peptides were used as crude products. For each peptide different concentrations were measured. Peptide 7 (*c*) and peptide 15 (*f*) did show the weakest signal. Peptide 11 (*d*) and 12 (*e*) showed similar signals, while peptide 16 (*g*) showed the highest signal. SPR: surface plasmon resonance.

3. The use of RED-tis-NTA dye for the MST requires a standard pre-test, which determines the affinity of the dye to the his-tag of the protein. This value should be reported, because the entire experimental design afterwards depends on this number. Also, for the optimal MST window I recommend to label the peptides with the fluorophore like Cy5 or disulfo-Cy5 and use these in the MST assay. I am aware of the fact that ADAMTS variant are not simple to handle and are very precious. However, the data would be most likely much more reliable. Currently, the MST amplitude below 2 provides a hint about the K_d , but no reliable value.

We thank the reviewer for these suggestions. We used the Cy5-labeled peptides for the new experiments. Each measurement is a merged sample set of three independent biological replicates. When we made the measurements with the originally found peptide probe and ADAMTS4, we could measure a ligand-induced fluorescence change. We tested this with a denaturation test, where we added 10 M urea to the three highest and lowest ADAMTS4 concentrations and after careful centrifugation and heating to 95 °C for 5 minutes, the fluorescence differences were gone. By using the ligand-induced fluorescence change, we got a binding constant of 51 ± 21 nM (Response to the reviewer figure 6). We can see a beginning binding against ADAMTS1 and ADAMTS5. Since we could not reach the plateau, a defined binding constant could be determined. From the data, we suggest a binding of around 1 μ M. By exchanging the glycine in the cycle against alanine, the binding affinity decrease to 0.6 ± 0.4 μ M, while it decreases even further by the exchange of serine with alanine. The exchange of arginine leads to a total loss of binding. This is consistent with the mutational computation study, in which we mutated both of the glutamic acids with glutamine and loss the binding, too. Our negative control peptide, the peptide 15 from the screening, which only showed a weak binding in SPR and a low signal in the screening did not show any binding in the MST.

An according paragraph was added to the manuscript.

Response to reviewer figure 6: By using the system suggested by the reviewer, we could substantially improve the MST measurements. We see a strong binding of our system against ADAMTS4, while the negative control peptide did not show a binding. This is consistent with the measured SPR data.

4. Careful observation of the MST data suggests that the same sample was measured three times. I recommend three biological replicates instead.

We apologize that our measurement approach was not made clear in the manuscript. We now clarified that each spectrum acquired in this study was a merged spectrum of three biological replicates, not technical replicates. An according paragraph describing this approach was added to the manuscript. For all novel MST measurements in this revision, analysis was performed using three biological replicates.

“Each measurement was repeated three times independently and the binding constant was calculated by merging all three measurements.”

5. Significant revision of the SPR and MST data/experiments is required to unambiguously demonstrate the nanomolar binding affinity and specificity of ADAMTS4 specific probe. Current data do not support this assumption with a doubt.

Based on the reviewer’s comments and suggestions, we adapted and improved the MST-measurements. Missing SPR data were added, as suggested. Since they were used for classification of the hits and were used with crude peptides, we do not want to use them for further and deeper revision. Therefore, the MST data can be used.

6. Fig 4b: only the minor portion of the staining with antibody and the probe overlap. Thus, this image is more indicative of lack of specificity than the proof of specificity. Please, adjust the interpretation accordingly.

We do agree with the reviewer that there is not a complete overlap between the stainings. This could be the result of different factors. While the fluorescent probe was administered *in vivo* via the tail vein. The ADAMTS4-antibody staining and microscopy was performed *ex vivo*. Therefore, the distribution and access of the probe and the antibody to the binding sites might slightly differ. We added an according statement to the manuscript. To support the assumption that ADAMTS-4 is expressed by CD68 macrophages and smooth muscle cells, we performed another *ex vivo* staining (Response-to-Reviewers Figure 7), which was added to the supplementary methods.

Response-to-Reviewers Figure 7: Expression of ADAMTS4 (b) by SMCs (a) and macrophages (c). *Ex vivo* immunofluorescence images of subsequent AAA slides following 4 weeks of Ang II infusion demonstrate the colocalization of ADAMTS4 with SMCs and CD68-positive cells.

7. *in vivo* specificity of the probe - using the same probe in "hot" and "cold" version does not underscore the specificity of the probe. It is expected that they will bind to the same binding crevice. We agree with the reviewer, that the competition between the "hot" and "cold" version of the peptide is not a test for specificity. This experiment was performed to demonstrate that no relevant unspecific binding of the probe in the aortic wall occurs. For analysis of specificity, we used the MST data of the comparison of ADAMTS4 to ADAMTS1 and ADAMTS5, in which we demonstrated a higher binding to ADAMTS4 (Response to the reviewer figure 8).

Additionally, docking experiment were performed and confirmed a single possible binding cleft. By mutation of the glutamic acids to glutamine in ADAMTS4, no binding of our probe could be measured. The same was observed by exchange of the arginine in the peptide with alanine (Response to the reviewer figure 9).

We adapted the according paragraphs in the manuscript to make this clearer to the reader.

Response-to-Reviewers Figure 8: In this MST experiment, ADAMTS1 and ADAMTS5 were used in the same way like ADAMTS4 with the Cy5 labeled peptide. The concentrations of the peptide was kept constant (3 nM in the measurement), while the proteins were diluted in the same manner as for ADAMTS4. A beginning binding was detected, but the plateau was not reached.

Response to the reviewer figure 9: Simulation snapshots of the in silico double mutation (Glu401Gln, Glu402/Gln) at early simulation stage (A, ~2 ns) and upon complete dissociation of the peptide (B, ~25 ns). Human AdamTS4 in orange cartoon, mutated residues as sticks, peptide as stick model including hydrogens.

8. In vivo MR imaging - statistical analysis of the data is needed to support the statements.

As suggested by the reviewers we added a thorough statistical analysis of the data to support the statements made in the study.

General observation:

when reading the manuscript the writing stiles between different part of the manuscript vary significantly. Some are easy to understand, in some improvements are recommended.

The complete manuscript was edited by a native speaker to improve readability and harmonize the writing styles between the different parts of the manuscript.

Reviewer #3 Comments highlighted in pdf file:

Page 6: estimated 75% of the full-length peptide could be presumed after the four truncation coupling steps

1. "What this assumption analytically confirmed?"

The assumption resulted from a mathematical estimation from the ratio of 93% Fmoc to 7% Boc aa. We could measure smaller truncation signals for later amino acids than for the early amino acids in the truncation system. Since this is amino acid depending, it is not appropriate for quality control.

First, the alkylation of the free thiol moiety in the truncation peptides with iodoacetamide, the sequences could be measured with MALDI-TOF-MS.

2. "This sentence is unfortunately difficult to understand. Please rephrase."

The sentence was rephrased as suggested.

We than alkylate the free thiol moiety in the truncation peptides with iodoacetamide, which reveals the truncation sequences in MALDI-TOF-MS.

Page 7: 22.000 beads from a 300.000 beads

3. "How many copies of the peptide you expect on a single bead?"

Rapp-Polymer indicates a capacity/bead [nmol] for TentaGel S with a capacity of 0.28 mmol/g and a size of 90 μm with 0.1 nmol/bead. Since our beads have a size of 75 μm and a capacity of 0.47 mmol/g, our beads should have a capacity of 0.1 nmol/bead (since the capacity is 1.68 times higher, but the volume is 1.73 times smaller). That means 58×10^{12} copies of peptides per bead, with 43.7×10^{12} copies of estimated full-length peptide. (<http://www.rapp-polymere.com/index.php?id=1219¤cy=eur>)

which was repeatedly regenerated for multi

4. "How have you confirmed that the chip was fully regenerated?"

During the validation of this technique in our group, we incubated the chip several times and after regeneration we measured the fluorescence and compared them. We did not detect significant differences in both scans. This indicated, that we have a nearly fully regeneration. For this chip, we measured only the fluorescence after the last regeneration. This scan was lower compared to the prescan, the ADAMTS4 and the ADAMTS5 scan, which is again a sign, that no protein was left

on the beads. Additionally, we could not find any traces of proteins in the MALDI-TOF-MS spectra.

We identified one candidate as the most promising binding peptide for ADAMTS4 with the sequence Ser-Cys-Gly-Arg-Arg-Ser-Cys* (Supp. Fig. 1b, c).*

5. *“The representation of SPR data needs to be adjusted, additional experimental details have to be added to the Supplemental figures. Please, see the comments in Supplemental for details.”*

We apologize to the reviewer for this oversight. All data are now added to the supplemental material. SPR was only use for classification to determine whether the peptides represent potential binders.

Page 8: For the resulting ADAMTS4- specific probe, we measured a binding constant of 38 nM (KD-confidence interval 11- 126 nM) against ADAMTS4 by microscale thermophoresis (MST),

6. *“Currently, the data from MST do not allow reliable determination of Kd. The data quality if unfortunately too poor (the amplitude too low; minimal recommended amplitude is 3). Also, why were the FAM-5 labeled peptides used in the combination with the RED-tris-NTA labeled protein? The MST experiments could be performed directly with FAM-5 labeled peptides or the peptides could be labeled directly with a red emitting dye like Cy5.”*

We thank the reviewer for his suggestion. We performed the scan like he mentioned with the Cy5 labeled peptide and the data were clearly improved.

Additionally, we measured the same probe against ADAMTS1 and ADAMTS5 in the same concentration range, and we could not detect binding in this range (Supp. Fig. 2a). Furthermore, by performing an alanine scan, the binding disappeared (Supp. Fig. 2b).

7. *“MST conditions were suboptimal. Thus, failing to detect binding might be just a coincidence.”*

By using the suggested measurement strategy of the reviewer, we could see a starting binding for ADAMTS1 and ADAMTS5. Both measurements did not reach the plateau, so a determination of the binding constant is not possible. Since we used directly the supplied concentration from both proteins we were limited for higher concentrations. We saw a loss of binding by trying to further concentrate the proteins, which might be caused due to the low stability of all three ADAMTS proteins.

As a negative control, we used the peptide sequence from the screening, which was the least binding candidate in the SPR experiment with the sequence Ser-Cys-Tyr-Pro-His-Phe-Cys*.*

8. *“The sentence is unfortunately not clear. Please rephrase.”*

We rephrased the sentence as suggested:

As a negative control, we used one of the peptides from the screening, which showed not only a weak signal against ADAMTS4 in the screening, but also only a very weak interaction in the SPR experiment. The peptide with the sequence Ser-Cys*-Tyr-Pro-His-Phe-Cys* did not show a binding in the ligand dependent fluorescence change (Fig. 2b).

Page 9

Only at increasing concentrations, we

9. *“At which concentrations?”*

We measured a decrease of cell activity with an MRI probe of 8.8 μmol for the 3 h cell test. By using statistical analysis of ANOVA and Sheffé Post Hoc tests, we did not measure a statistically significance with the negative control. We performed additionally CytoTox-ONE and Celltiter-Glo tests with both cell lines and the same concentrations of the probe and timepoints. In this test we did not measure a statistically significance to the negative control.

minor inhibition / small influence on cell activity

10. *“Was the difference statistically significant?”*

The differences were not statistically significant. We changed it accordingly in the manuscript.

Page 10

significant increase / no significant increase

11. *“Please add statistical parameters“*

As suggested the according statistical parameters were added.

Page 11: Both showed fluorescence in similar regions, which supports the specificity of the novel probe. 12. *“This statement does actually imply that there was a difference in the regions stained with antibody and the probe. In fact, close observation of images shows the overlap only in very specific region. This is indicative of lack of specificity for one of the probing tools.“*

Please refer to the main response answer question 6.

in vivo specificity of the ADAMTS4-specific probe

13. *“The suppression of the binding of the "hot" probe by the same "cold" probe is not a proof of specificity. Both probes namely bind to the same target(s) and it is thus logical that the "cold" probe will suppress the binding of identical "hot" probe.“*

Please refer to the main response answer question 7.

A weak signal for this peptide was found in the screening but did not show binding in the SPR-preanalysis as well as in the MST measurement.

14. *“The data from biophysical characterization of these peptides are unfortunately noninclusive.”*

We apologize this and add all the SPR data to the SI.

Page 25: After removing the mixture, the beads were washed three times each with ultrapure water, DMF, DCM, and ether and dried under vacuum.

15. *“How was the library QCed?”*

Please see answer to question 1 in the main response.

Page 27: stable baseline was reached.

16. "What was the level of the protein capture?"

By using the equation

$$R_{max} = \frac{R_{anti-His-Ab} * M_{ADAMTS4} * Valency_{anti-His-Ab}}{M_{anti-His-Ab}}$$

we calculated a R_{max} of 5524 μ RIU. After immobilization of ADAMTS4 and flushing for around 5 $\frac{1}{2}$ h, we had an R-value of 1469 μ RIU (for the difference), which complies to around 26% of the maximum was captured.

Before and after the run, a blank was measured.

17. "Have you observed and experimentally checked for any unspecific interactions of the peptides with the chip covered with Anti-His-Ab?"

Since both channels were covered with Anti-His-Ab, unspecific interactions between the peptides and the antibody will be cancelled out by calculating the difference of the sample and the reference channel.

Page 28: The sample was diluted with the MST Buffer (PBS-Brij-35 (0,03%), pH 7.3, filtered with 0.2 μ m) 1:5 to achieve an end concentration of 10 nM dye.

18. "Was the concentration of the protein at this point 20 nM? Under these conditions most likely some free dye will be present in the assay mixture, which will cause the reduction of the observed amplitude. What was the affinity of the dye for the His-tag of the investigated proteins?"

The protein concentration was indeed 20 nM. Based on the comments and suggestions of the reviewer we changed the system.

FAM-5 labeled peptides

19. "Why haven't you directly used these peptides in MST and leave the protein unlabeled? Also, the peptides could be conjugated to the red emitting dye. Using the fluorescence of the labeled peptide as MST readout could improve the data quality significantly."

As suggested the peptides were now labeled to Cy5 and used with the unlabeled proteins for the MST measurement.

Page 40: Statistical Analysis

20. „Add the statistic evaluation of either experiments too.”

The statistical parameters were added as suggested.

Page 60: followed by the ADAMTS4.

21. „How much of the protein was captured? Add the sensogram showing the protein capture and documenting potential basal line drift.”

We added the sensogram and calculated the protein capture (please refer to question 16).

For the peptide 11, 12, and 16, lower concentrations had to be measured, due to the high signal response.

22. "Add the sensograms from each specific interaction between ADAMTS4 and given peptide (RU vs. time)."

We added the sensograms in the supplementary material.

After the measurement of all five peptides, peptide number 16 showed the highest signal at the lowest concentration.

"Is the interaction specific? Do you observe any unspecific binding to the chip in the absence of ADAMTS?"

Please refer to question 17.

Reviewer #4 (Remarks to the Author):

This paper by Kaufmann et al. describes the development of an ADAMTS4-specific MR probe, which they then tested to visualize AAA in an ApoE^{-/-}+Ang II model.

Strength: A marker for early detection of AAA or TAA is one of the deficiencies in the medical field, therefore, this study is a step in the right direction.

We thank the reviewer for the appreciation of our study.

Weaknesses:

1) When are ADAMTS4 levels elevated in an aneurysmal aorta? How early before the development of AAA?

This is a highly important point. To address these questions, we performed an additional set of experiments. This included an additional group of ApoE^{-/-} mice implanted with angiotensin loaded minipumps (n=20), which were investigated in a longitudinal study setup.

Regarding the timeline, the mice underwent a native MRI followed by an ADAMTS4-MRI at day 2, 4 and 10 after AngII infusion. At day 28 post implantation, only native imaging was performed to assess whether a AAA developed. Subsequently the aortic samples were excised (Response to the reviewer figure 10).

Response to the reviewer figure 10: Set-up of longitudinal study at ADAMTS-4 in vivo MRI at day 2, 4 and 10 after ATII-infusion. At day 28 post implantation, native imaging was performed to assess whether the AAA has developed.

In this longitudinal study, setup we could demonstrate that the ADAMTS4-MRI level in the aortic wall significantly increases 4 days after the implantation of the ATII-minipumps (Response to the reviewer figure 11). Only a minor further increase can be measured after 10 days.

More importantly, we could demonstrate that animals with the highest increase in ADAMTS4-MRI (CNR: 15.2 ± 0.7 , Response to the reviewer figure 11, c) were most likely to suffer a fatal AAA rupture. Animals with a moderate increase in ADAMTS4-MRI levels (CNR: 10.3 ± 1.1 , Response to the reviewer figure 11, b) which developed an AAA however survived for four weeks. Animals with no significant increase in ADAMTS4 did not develop an AAA (CNR: 6.9 ± 1.2 , Response to the reviewer figure 11, a). The 4 day timepoint therefore was the most promising early timepoint for the prediction of an AAA development and survival.

We added an according paragraph and the additional figures to the manuscript.

Response to the reviewer figure 11: **a**, The group that did not develop an AAA did not show a significant increase in ADAMTS4-MRI levels two, four or 10 days after the initiation of ATII-infusion. **b**, the group that developed an AAA however survived for four weeks showed a moderate increase of ADAMTS4-MRI levels after 4 days with no further significant increase after 10 days. **c**, The fatal AAA rupture group showed the strongest increase in ADAMTS4-MRI levels of the 4 days, with no further significant increase after 10 days. **d**, ADAMTS4-MRI levels measured after 4 days significantly increased in the group that developed in AAA and survived for four weeks, compared to the group that did not develop any AAA. The group that suffered a fatal AAA rupture showed further significant increase ADAMTS4-MRI levels at this time point. The 4 day timepoint therefore seems to be the most promising timepoint for the prediction of an AAA development and survival.

2) Related to the previous point, the authors show that ADAMTS4-MR probe labeled the aneurysmal aorta (at 2wk and 4wks), however in both time points there is a visible AAA which is also detected by MRI or echo-ultrasound (echo would give more clear images than the shown MRI images), therefore, using this probe would not really help in early detection of the AAA.

We fully agree that the aortic diameter or aortic area is currently the only clinic established parameter for the assessment were AAAs. However, novel biomarkers for the evaluation of risk of aortic rupture are urgently needed.¹

Based on the longitudinal study setup described above, we could demonstrate that the ADAMTS4-MRI level is a stronger predictor for the development of a fatal AAA rupture (area under the curve: 0.96), compared to the aortic area (area under the curve: 0.78, Response to the reviewer figure 12). In the clinical setting, there is common consensus that patients with AAAs larger than 55 mm should undergo surgery². There is, however, still controversy surrounding the management of asymptomatic medium-sized AAAs (40–55 mm)^{2,3}. No biomarker is currently available, which could help to better characterize aortic aneurysms in patients and to predict their risk of rupture.^{2,3} The ADAMTS4-MRI level, as a novel biomarker, could be especially important for guiding therapeutic decisions regarding these aortic aneurysms between 40–55 mm.

We added an according paragraph and the additional figures to the manuscript.

Response to the reviewer figure 12: **a**, The ROC analysis demonstrates that the ADAMTS4-MRI level measured after 4 days enables of the prediction of a fatal AAA rupture with a sensitivity of 0.88 and a specificity of 0.9 (ROC Curve Area 0.96, Standard Error 0.04, 95% Confidence Interval 0.90 To 1.04, $P < 0.05$). **b**, The ROC analysis demonstrates that the aortic area measured after 4 days enables of the prediction of a fatal AAA rupture with a sensitivity of 0.78 and a specificity of 0.76 (ROC Curve Area 0.78, Standard Error 0.11, 95% Confidence Interval 0.55 To 1.01, $P < 0.05$).

3) It is also important to know which cell types in the aorta express ADAMTS4 when undergoing AAA development, this would further help in identifying the type of aneurysm. E.g. the ApoE^{-/-}+AngII model (used in this study) causes an outward remodeling where the outer diameter of AAA is greater than the lumen dilation (and is associated with plaque deposition and metabolic alterations).

It has been demonstrated that in a murine model of AAA, ADAMTS4 is expressed by smooth muscle cells in the aortic wall.⁴ In addition, ADAMTS4 can also be expressed by endothelial cells and macrophages.⁵ To the best of our knowledge, the ApoE^{-/-} + AngII murine model develops fusiform, infrarenal aortic aneurysms, usually accompanied by the formation of an intramural thrombus. To support the assumption that ADAMTS4 is expressed by CD68 macrophages and

smooth muscle cells, we performed another ex vivo staining (Response-to-Reviewers Figure 13), which was added to the supplementary methods of our manuscript.

Response-to-Reviewers Figure 13: Expression of ADAMTS4 (b) by SMCs (a) and macrophages (c). Ex vivo immunofluorescence images of subsequent AAA slides following 4 weeks of Ang II infusion demonstrate the colocalization of ADAMTS4 with SMCs and CD68-positive cells.

Would ADAMTS4 probe labeling also work where there is aortic wall degradation and lumen dilation without the outward remodeling?

The data from the longitudinal study indicate that there is an early increase in the ADAMTS-4 MRI level after 4 days. In a subgroup of animals (n=4) we observed a strong increase in ADAMTS-4 levels (CNR:11.7±0.8) while no significant outward remodeling was measured on MRI. However, at the later time points after 10 days and 28 days a significant outward remodeling was observed. This indicates that the ADAMTS-4 expression precedes the outward remodeling in the aortic wall.

Reviewer #5 (Remarks to the Author):

This manuscript reports the identification of a novel cyclic peptide with a high affinity to ADAMTS4 through screening of a one-bead-one-compound library. The peptide was then modified for development of an MR probe through conjugation of DOTA-gadolinium. Proof-of-concept validations determined the potential of this probe for detection of abdominal aortic aneurysm using a mouse model. Molecular imaging of key processes involved in the pathogenesis of aortic aneurysm is a highly needed strategy to improve the management of patients. The approach and targeted molecule are novel, and the experimental approach is robust. I only have a few comments, as summarized below.

We thank the reviewer for the assessment of our manuscript.

1. Figure 2b: There is a marked variation in % binding both within each concentration and between different concentrations. It is also unclear if the binding has reached the plateau. These affect K_d measurement. Increasing N may help to resolve the issue. The same issues apply to Supplemental Figure 2.

We agree with the reviewer. In accordance with the suggestions of Reviewer 3, we used the Cy5-labeled peptides and diluted the proteins for the MST measurements and could significantly improve the MST measurements (Response to the reviewer figure 14). Nevertheless, reaching the plateau in this case is limited by the concentration of the proteins. Several attempts to increase the concentration by protein concentrators lead to a loss of binding. Since ADAMTS4/5/1 enzymes are highly unstable, their structure is compromised by centrifugation. For our system and also for the alanine exchange of the glycine, we reached nearly the plateau, that is why we can determine a binding constant. For ADAMTS5 and ADAMTS1 and the alanine exchange of the serine, we can see a starting curve, but no binding constant can be determined. All other three measurements did not show binding.

Response to reviewer figure 14: By using the system suggested by the reviewer, we could substantially improve the MST measurements. We see a strong binding of our system against ADAMTS4, while the negative control peptide did not show a binding. This is consistent with the measured SPR data.

2. Figure 4c: Representative images from mice with blocking experiments are required.

As suggested, we added representative images from mice with blocking experiments to Figure 4c (Response to the reviewer figure 15).

Response to the reviewer figure 15: The competition experiment showed a significant decrease in the contrast-to-noise-ratio compared to the administration of the gadolinium-labeled ADAMTS4-specific probe alone, confirming the specific binding of the probe.

3. Please explain the rationale for using a sevenfold dose of non-labeled probe in blocking experiments. This dose seems very low to achieve saturation of the target molecule (ADAMTS4) in tissues. Therefore, it is surprising that ~80-90% blocking is achieved with this dose. Biodistribution data from blocking experiments will also be helpful.

We agree with the reviewer, that a sevenfold dose might seem to be very low. But in our case, we use a contrast agent amount, which complies with the clinical dose of an unspecific contrast agent. By using this amount, we see after application of the contrast agent a strong accumulation of the contrast agent in the kidneys, implying that the amount of probe is very high. So the sevenfold dose of the cold probe should be enough for blocking most of the accessible ADAMTS4 targets.

In this context, mutation studies are of high importance to validate the binding-mechanism. We performed a MD-simulation to address this issue. The simulation demonstrated a fast peptide break from the position by mutation of one of the glutamine acids in the binding cleft. This is in line with our hypothesis regarding the binding cleft and also the exchange of arginine by alanine in the MST experiments.

We adapted the manuscript accordingly and added the additional figure (Response to the reviewer figure 16,17).

Response to the reviewer figure 16: Simulation snapshots of the *in silico* double mutation (Glu401Gln, Glu402Gln) at early simulation stage (A, ~2 ns) and upon complete dissociation of the peptide (B, ~25 ns). Human AdamTS4 in orange cartoon, mutated residues as sticks, peptide as stick model including hydrogens.

Response to the reviewer figure 17: Simulation repetition and mutation of ADAMTS4. a, Root-mean-square deviation (RMSD of non-hydrogen atoms) vs. simulation time for three separate simulations of ADAMTS4 in complex with peptide binder (A-C). Simulations were started with different initial velocity distributions. The RMSD of the complete protein vs. simulation time is indicated in black, the RMSD of the peptide binder with respect to the start placement (after best superposition of the protein on the start structure) is shown in red. (D) same simulation for the mouse AdamTS in complex with the peptide binder (same initial position as in human ADAMTS4 cases). (E) same as in (A) but with the *in silico* human ADAMTS4 Glu401Gln, Glu402Gln mutation. Note, the RMSD scale is different in (E).

4. Figure 4d-e: The increase in CNR after injection of the negative control probe seems visually a lot more than the quantified values in panel D. Perhaps panel E may be replaced by a more representative image. Also, it is ambiguous to which groups the reported P values refer to (panel D).

We thank the reviewer for this comment. As suggested the figure was adapted to give the reader a clearer representation of the measured CNR-values between panel D and E (Response to the reviewer figure 18).

We also apologize for the oversight with the misaligned P-values in the figure. This was corrected to make clear that the P-values refer to the comparison between pre-contrast and ADAMTS4 scans.

Response to the reviewer figure 18: Using the negative control probe *Cys*-Phe-His-Pro-Tyr-Cys*-linker (DOTA-Gd)* with a similar size to the ADAMTS4-specific probe, no significant increase in contrast-to-noise ratio (CNR) was measured in vivo in the aneurysmatic wall of *ApoE-/-* mice. The application of the ADAMTS4-specific probe resulted in a strong and significant increase in CNR in the vessel wall as well in the thrombus.

References Response to Reviewers

1. Bengtsson, H., Sonesson, B. & Bergqvist, D. Incidence and prevalence of abdominal aortic aneurysms, estimated by necropsy studies and population screening by ultrasound. *Ann N Y Acad Sci* **800**, 1-24 (1996).
2. Sakalihasan, N., Limet, R. & Defawe, O.D. Abdominal aortic aneurysm. *Lancet* **365**, 1577-1589 (2005).
3. Lederle, F.A., *et al.* Immediate repair compared with surveillance of small abdominal aortic aneurysms. *N. Engl. J. Med.* **346**, 1437-1444 (2002).
4. Ren, P., *et al.* Critical Role of ADAMTS-4 in the Development of Sporadic Aortic Aneurysm and Dissection in Mice. *Sci. Rep.* **7**, 12351 (2017).
5. Dong, H., *et al.* Relationship between ADAMTS4 and carotid atherosclerotic plaque vulnerability in humans. *J. Vasc. Surg.* **67**, 1120-1126 (2018).
6. Trachet, B., *et al.* Angiotensin II infusion into ApoE^{-/-} mice: a model for aortic dissection rather than abdominal aortic aneurysm? *Cardiovasc. Res.* **113**, 1230-1242 (2017).
7. Golledge, J. & Norman, P.E. Atherosclerosis and abdominal aortic aneurysm: cause, response, or common risk factors? *Arterioscler. Thromb. Vasc. Biol.* **30**, 1075-1077 (2010).

Reviewers' Comments:

Reviewer #1:

Remarks to the Author:

I would like to thank authors for covering almost every aspect I have mentioned in detail.

Reviewer #2:

Remarks to the Author:

As my comments have been acceptable revised.

Reviewer #3:

Remarks to the Author:

The authors applied innovative OBOC approach to synthesize the library of cyclic peptide with the intention to identify the specific binder for ADAMTS4. Assumed specific probe was then used in a series of in vivo experiments to demonstrate its utilization in the AAA field. The lack of specific probe namely does not allow timely diagnosis of potentially fatal condition. The work is of significance to the OBOC field.

I thank to the authors to address the comments in a detailed manner and to greatly improve the MST data quality. These data now fully support the conclusions stated by the authors.

Reviewer #4:

Remarks to the Author:

The authors have addressed my concerns.

Reviewer #5:

Remarks to the Author:

The authors have been very responsive and have adequately addressed the previous comments. I have no additional comment.

Final point-by-point response to these comments NCOMMS-21-00839A-Z
All changes in the manuscript text file are also indicated by colour highlighting.

Reviewer #1 (Remarks to the Author):

I would like to thank authors for covering almost every aspect I have mentioned in detail.

We thank the reviewer for his comment and for his concern in the previous revision, which helped a lot to improve our manuscript.

Reviewer #2 (Remarks to the Author):

As my comments have been acceptable revised.

We thank the reviewer for his comment and also for his previous comments, which improved our manuscript.

Reviewer #3 (Remarks to the Author):

The authors applied innovative OBOC approach to synthesize the library of cyclic peptide with the intention to identify the specific binder for ADAMTS4. Assumed specific probe was then used in a series of in vivo experiments to demonstrate its utilization in the AAA field. The lack of specific probe namely does not allow timely diagnosis of potentially fatal condition. The work is of significance to the OBOC field.

I thank to the authors to address the comments in a detailed manner and to greatly improve the MST data quality. These data now fully support the conclusions stated by the authors.

We thank the reviewer for the appreciation of our study. We want to thank him again for all his previous hints, which helped us a lot to greatly improve our manuscript.

Reviewer #4 (Remarks to the Author):

The authors have addressed my concerns.

We thank the reviewer for his comment and also for his questions from the first revision, which helped us to increase the value of our thesis.

Reviewer #5 (Remarks to the Author):

The authors have been very responsive and have adequately addressed the previous comments. I have no additional comment.

We thank the reviewer for the appreciation of our study and for all his previous comments. They helped us, to improve the manuscript.